# Learning to Accelerate Partial Differential Equations via Latent Global Evolution

**Tailin Wu**
Department of Computer Science
Stanford University
tailin@cs.stanford.edu

**Takashi Maruyama**
NEC Corp. & Stanford University
49takashi@nec.com &
takashi279@cs.stanford.edu

**Jure Leskovec**
Department of Computer Science
Stanford University
jure@cs.stanford.edu

## Abstract

Simulating the time evolution of Partial Differential Equations (PDEs) of large-scale systems is crucial in many scientific and engineering domains such as fluid dynamics, weather forecasting and their inverse optimization problems. However, both classical solvers and recent deep learning-based surrogate models are typically extremely computationally intensive, because of their local evolution: they need to update the state of each discretized cell at each time step during inference. Here we develop Latent Evolution of PDEs (LE-PDE), a simple, fast and scalable method to accelerate the simulation and inverse optimization of PDEs. LE-PDE learns a compact, global representation of the system and efficiently evolves it fully in the latent space with learned evolution models. LE-PDE achieves speed-up by having a much smaller latent dimension to update during long rollout as compared to updating in the input space. We introduce new learning objectives to effectively learn such latent dynamics to ensure long-term stability. We further introduce techniques for speeding up inverse optimization of boundary conditions for PDEs via backpropagation through time in latent space, and an annealing technique to address the non-differentiability and sparse interaction of boundary conditions. We test our method in a 1D benchmark of nonlinear PDEs, 2D Navier-Stokes flows into turbulent phase and an inverse optimization of boundary conditions in 2D Navier-Stokes flow. Compared to other strong baselines, we demonstrate up to $128\times$ reduction in the dimensions to update, and up to $15\times$ improvement in speed, while achieving competitive accuracy. [1].

## 1 Introduction

Many problems across science and engineering are described by Partial Differential Equations (PDEs). Among them, temporal PDEs are of huge importance. They describe how the state of a (complex) system evolves with time, and numerically evolving such equations are used for forward prediction and inverse optimization across many disciplines. Example application includes weather forecasting [1], jet engine design [2], nuclear fusion [3], laser-plasma interaction [4], astronomical simulation [5], and molecular modeling [6].

To numerically evolve such PDEs, decades of works have yielded (classical) PDE solvers that are tailored to each specific problem domain [7]. Albeit principled and accurate, classical PDE solvers

---

[1]Project website and code can be found at `http://snap.stanford.edu/le_pde/`.

36th Conference on Neural Information Processing Systems (NeurIPS 2022).

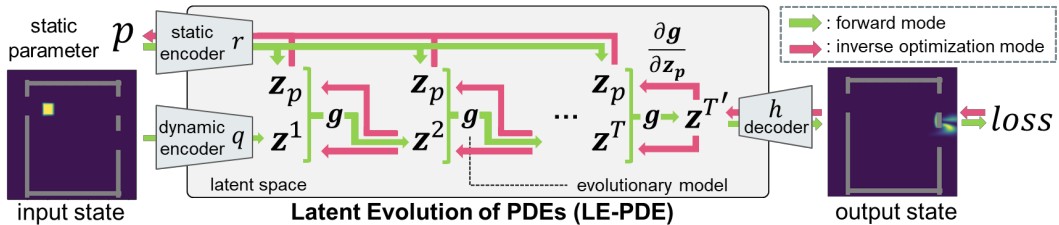

Figure 1: LE-PDE schematic. In forward mode (green), LE-PDE evolves the dynamics in a global latent space. In inverse optimization mode (red), it optimizes parameter $p$ (*e.g.* boundary) through latent unrolling. The compressed latent vector and dynamics can significantly speed up both modes.

are typically slow due to the small time steps or implicit method required for numerical stability, and their time complexity typically scales linearly or super-linearly with the number of cells the domain is discretized into [8]. For practical problems in science and engineering, the number of cells at each time step can easily go up to millions or billions and may even require massively parallel supercomputing resources [9, 10]. Besides forward modeling, inverse problems, such as inverse optimization of system parameters and inverse parameter inference, also share similar scaling challenge [11]. How to effectively speed up the simulation while maintaining reasonable accuracy remains an important open problem.

Recently, deep learning-based surrogate models have emerged as attractive alternative to complement [12] or replace classical solvers [13, 14]. They directly learn the dynamics from data and alleviate much engineering effort. They typically offer speed-up due to explicit forward mapping [15, 16], larger time intervals [14], or modeling on a coarser grid [12, 17]. However, their evolution scales with the discretization, since they typically need to update the state of each discretized cell at each time step, due to the local nature of PDEs [18]. For example, if a problem is discretized into 1 million cells, deep learning-based surrogate models (*e.g.*, CNNs, Graph Networks, Neural Operators) will need to evolve these 1 million cells per time step. How to go beyond updating each individual cells and further speed up such models remains a challenge.

Here we present Latent Evolution of PDEs (LE-PDE) (Fig. 1), a simple, fast and scalable method to accelerate the simulation and inverse optimization of PDEs. Our key insight is that a common feature of the dynamics of many systems of interest is the presence of dominant, low-dimensional coherent structures, suggesting the possibility of efficiently evolving the system in a low-dimensional global latent space. Based on this observation, we develop LE-PDE, which learns the evolution of dynamics in a global latent space. Here by "global" we mean that the dimension of the latent state is fixed, instead of scaling linearly with the number of cells as in local models. LE-PDE consists of a dynamic encoder that compresses the input state into a dynamic latent vector, a static encoder that encodes boundary conditions and equation parameters into a static latent vector, and a latent evolution model that evolves the dynamic latent vector fully in the latent space, and decode via a decoder only as needed. Although the idea of latent evolution has appeared in other domains, such as in computer vision [19, 20, 21] and robotics [22, 23, 24, 25], these domains typically have clear object structure in visual inputs allowing compact representation. PDEs, on the other hand, model dynamics of continuum (*e.g.*, fluids, materials) with infinite dimensions, without a clear object structure, and sometimes with chaotic turbulent dynamics, and it is pivotal to model their long-term evolution accurately. Thus, learning the latent dynamics of PDEs presents unique challenges.

We introduce a multi-step latent consistency objective, to encourage learning more stable long-term evolution in latent space. Together with the multi-step loss in the input space, they encourage more accurate long-term prediction. To accelerate inverse optimization of PDEs which is pivotal in engineering (*e.g.* optimize the boundary condition so that the evolution of the dynamics optimizes certain predefined objective), we show that LE-PDE can allow faster optimization, via backpropagation through time in latent space instead of in input space. To address the challenge that the boundary condition may be non-differentiable or too sparse to receive any gradient, we design an annealing technique for the boundary mask during inverse optimization.

We demonstrate our LE-PDE in standard PDE-learning benchmarks of a 1D family of nonlinear PDEs and a 2D Navier-Stokes flow into turbulent phase, and design an inverse optimization problem in 2D Navier-Stokes flow to probe its capability. Compared with state-of-the-art deep learning-based

surrogate models and other strong baselines, we show up to $128\times$ reduction in the dimensions to update and up to $15\times$ speed-up compared to modeling in input space, and competitive accuracy.

## 2 Problem Setting and Related Work

We consider temporal Partial Differential Equations (PDEs) w.r.t. time $t \in [0, T]$ and multiple spatial dimensions $\mathbf{x} = [x_1, x_2, ...x_D] \in \mathbb{X} \subseteq \mathbb{R}^D$. We follow similar notation as in [7]. Specifically,

$$\partial_t \mathbf{u} = F(\mathbf{x}, \mathbf{a}, \mathbf{u}, \partial_{\mathbf{x}}\mathbf{u}, \partial_{\mathbf{xx}}\mathbf{u}, ...), \qquad (t, \mathbf{x}) \in [0, T] \times \mathbb{X}, \qquad (1)$$

$$\mathbf{u}(0, x) = \mathbf{u}^0(\mathbf{x}), \ B[\mathbf{u}](t, \mathbf{x}) = 0, \qquad \mathbf{x} \in \mathbb{X}, (t, \mathbf{x}) \in [0, T] \times \partial\mathbb{X}. \qquad (2)$$

Here $\mathbf{u} : [0, T] \times \mathbb{X} \to \mathbb{R}^n$ is the solution, which is an infinite-dimensional function. $\mathbf{a}$ is time-independent static parameters of the system, which can be defined on each location $\mathbf{x}$, *e.g.* diffusion coefficient that varies in space but static in time, or a global parameter. $F$ is a linear or nonlinear function on the arguments of $(\mathbf{x}, \mathbf{a}, \mathbf{u}, \partial_{\mathbf{x}}\mathbf{u}, \partial_{\mathbf{xx}}\mathbf{u}, ...)$. Note that in this work we consider time-independent PDEs where $F$ does not explicitly depend on $t$. $\mathbf{u}^0(\mathbf{x})$ is the initial condition, and $B[\mathbf{u}](t, \mathbf{x}) = 0$ is the boundary condition when $\mathbf{x}$ is on the boundary of the domain $\partial\mathbb{X}$ across all time $t \in [0, T]$. Here $\partial_{\mathbf{x}}\mathbf{u} = \frac{\partial \mathbf{u}}{\partial \mathbf{x}}$, $\partial_{\mathbf{xx}}\mathbf{u} = \frac{\partial^2 \mathbf{u}}{\partial \mathbf{x}^2}$ are first- and second-order partial derivatives, which are a matrix and a 3-order tensor, respectively (since $\mathbf{x}$ is a vector). *Solving* such temporal PDEs means computing the state $\mathbf{u}(t, \mathbf{x})$ for any time $t \in [0, T]$ and location $\mathbf{x} \in \mathbb{X}$ given the above initial and boundary conditions.

**Classical solvers for solving PDEs**. To numerically solve the above PDEs, classical numerical solvers typically discretize the domain $\mathbb{X}$ into a finite grid or mesh $X = \{c_i\}_{i=1}^N$ with $N$ non-overlapping cells. Then the infinite-dimensional solution function of $\mathbf{u}(t, \mathbf{x})$ is discretized into $U^k = \{\mathbf{u}_i^k\}_{i=1}^N \in \mathbb{U}$ for each cell $i$ and time $t = t_k, k = 1, 2, ...K$. $\mathbf{a}$ is similarly discretized into $\{a_i\}_{i=1}^N$ with values in each cell. Mainstream numerical methods, including Finite Difference Method (FDM) and Finite Volume Method (FVM), proceed to evolve such temporal PDEs by solving the equation at state $\{\mathbf{u}_i^{k+1}\}$ at time $t = t_{k+1}$ from state $\{\mathbf{u}_i^k\}$ at time $t_k$. These solvers are typically slow due to small time/space intervals required for numerical stability, and needing to update each cell at each time steps. For more detailed information on classical solvers, see Appendix A.

**Deep learning-based surrogate modeling**. There are two main approaches in deep learning-based surrogate modeling. The first class of method is autoregressive methods, which learns the mapping $f_\theta$ with parameter $\theta$ of the discretized states $U^k$ between consecutive time steps $t_k$ and $t_{k+1}$: $\hat{U}^{k+1} = f_\theta(\hat{U}^k, p), k = 0, 1, 2, ....$ Here $\hat{U}^k = \{\hat{\mathbf{u}}_i^k\}_{i=1}^N$ is the model $f_\theta$'s predicted state for $U^k = \{\mathbf{u}_i^k\}_{i=1}^N$ at time $t_k$, with $\hat{U}^0 := U^0$. $p = (\partial\mathbb{X}, \{a_i\}_{i=1}^N)$ is the system parameter which includes the boundary domain $\partial\mathbb{X}$ and discretized static parameters $\{a_i\}_{i=1}^N$. Repetitively apply $f_\theta$ at inference time results in autoregressive rollout

$$\hat{U}^{k+m} = (f_\theta(\cdot, p))^{(m)} (\hat{U}^k), m \geq 1. \qquad (3)$$

Here $f_\theta(\cdot, p) : \mathbb{U} \to \mathbb{U}$ is a partial function whose second argument is fulfilled by the static system parameter $p$. Typically $f_\theta$ is modeled using CNNs (if the domain $\mathbb{X}$ is discretized into a grid), Graph Neural Networks (GNNs, if the domain $\mathbb{X}$ is discretized into a mesh). These methods all involve local computation, where the value $u_i^{k+1}$ at cell $i$ at time $t_{k+1}$ depend on its neighbors $\{u_j^k\}_{j \in \mathcal{N}(i)}$ at time $t_k$, where $\mathcal{N}(i)$ is the set of neighbors up to certain hops. Such formulation includes CNN-based models [26], GNN-based models [7, 27, 28] and their hierarchical counterparts [18, 29]. The surrogate modeling with local dynamics makes sense, since the underlying PDE is essentially a local equation that stipulates how the solution function $\mathbf{u}$'s value at location $\mathbf{x}$ depends on the values at its infinitesimal neighborhood. The second class of method is Neural Operators [14, 30, 31, 32, 33, 34, 35, 36], which learns a neural network (NN) that approximates a mapping between infinite-dimensional functions. Although having the advantage that the learned mapping is discretization invariant, given a specific discretization, Neural Operators still needs to update the state at each cell based on neighboring cells (and potentially cells far away), which is still inefficient at inference time, especially dealing with larger-scale problems. In contrast to the above classes of deep learning-based approaches that both requires local evolution at inference time, our LE-PDE method focuses on improving efficiency. Using a learned global latent space, LE-PDE removes the need for local evolution and can directly evolve the system dynamics via a global latent vectors $\mathbf{z}^k \in \mathbb{R}^{d_z}$ for time $t_k$. This offers great potential for speed-up due to the significant reduction in representation.

**Inverse optimization**. Inverse optimization is the problem of optimizing the parameters $p$ of the PDE, including boundary $\partial\mathbb{X}$ or static parameter $\mathbf{a}$ of the equation, so that a predefined objective $L_d[\mathbf{a}, \partial\mathbb{X}] = \int_{t=t_s}^{t_e} \ell_d[\mathbf{u}(t, \mathbf{x})]dt$ is minimized. Here the state $\mathbf{u}(t, \mathbf{x})$ implicitly depends on $\mathbf{a}, \partial\mathbb{X}$ through the PDE (Eq. 1) and the boundary condition (Eq. 2). Such problems have huge importance in engineering, *e.g.* in designing jet engines [2] and materials [37] where the objective can be minimizing drag or maximizing durability, and inverse parameter inference (*i.e.* history matching) [38, 39, 40] where the objective can be maximum a posteriori estimation. To solve such problem, classical methods include adjoint method [41, 42], shooting method [43], collocation method [44], etc. One recent work [45] explores optimization via backpropagation through differential physics in the input space, demonstrating speed-up and improved accuracy compared to classical CEM method [46]. However, for long rollout and large input size, the computation becomes intensive to the point of needing to save gradients in files. In comparison, LE-PDE allows backpropagation in latent space, and due to the much smaller latent dimension and evolution model, it can significantly reduce the time complexity in inverse optimization.

**Reduced-order modeling**. A related class of work is reduced-order modeling. Past efforts typically use linear projection into certain basis functions [47, 48, 49, 50, 51, 52, 53, 54] which may not have enough representation power. A few recent works explore NN-based encoding [55, 56, 57, 58, 59, 60] for fluid modeling. Compared to the above works, we focus on speeding up simulation and inverse optimization of more general PDEs using expressive NNs, with novel objectives, and demonstrate competitive performance compared to state-of-the-art deep learning-based models for PDEs.

## 3 Our approach LE-PDE

In this section, we detail our Latent Evolution of Partial Differential Equations (LE-PDE) method. We first introduce the model architecture (Sec. 3.1, and then we introduce learning objective to effectively learn faithfully long-term evolution (Sec. 3.2). In Sec. 3.3, we introduce efficient inverse optimization in latent space endowed by our method.

### 3.1 Model architecture

The model architecture of LE-PDE consists of four components: (1) a dynamic encoder $q : \mathbb{U} \rightarrow \mathbb{R}^{d_z}$ that maps the input state $U^k = \{\mathbf{u}_i^k\}_{i=1}^N \in \mathbb{U}$ to a latent vector $\mathbf{z}^k \in \mathbb{R}^{d_z}$; (2) an (optional) static encoder $r : \mathbb{P} \rightarrow \mathbb{R}^{d_{zp}}$ that maps the (optional) system parameter $p \in \mathbb{P}$ to a static latent embedding $\mathbf{z}_p$; (3) a decoder $h : \mathbb{R}^{d_z} \rightarrow \mathbb{U}$ that maps the latent vector $\mathbf{z}^k \in \mathbb{R}^{d_z}$ back to the input state $U^k$; (4) a latent evolution model $g : \mathbb{R}^{d_z} \times \mathbb{R}^{d_{zp}} \rightarrow \mathbb{R}^{d_z}$ that maps $\mathbf{z}^k \in \mathbb{R}^{d_z}$ at time $t_k$ and static latent embedding $\mathbf{z}_p \in \mathbb{R}^{d_{zp}}$ to $\mathbf{z}^{k+1} \in \mathbb{R}^{d_z}$ at time $t_{k+1}$. We employ the temporal bundling trick [7] where each input state $U^k$ can include states over a fixed length $S$ of consecutive time steps, in which case each latent vector $\mathbf{z}_k$ will encode such bundle of states, and each latent evolution will predict the latent vector for the next bundle of $S$ steps. $S$ is a hyperparameter and may be chosen depending on the problem, and $S = 1$ reduces to no bundling. A schematic of the model architecture and its inference is illustrated in Fig. 1. Importantly, we require that for the dynamic encoder $q$, it needs to have a flatten operation and MultiLayer Perception (MLP) head that maps the feature map into a single fixed-length vector $\mathbf{z} \in R^{d_z}$. In this way, the dimension of the latent space does not scale linearly with the dimension of the input, which has the potential to significantly compress the input, and can make the long-term prediction much more efficient. At inference time, LE-PDE performs autoregressive rollout in latent space $\mathbb{R}^{d_z}$:

$$\hat{U}^{k+m} = h \circ g\left(\cdot, r(p)\right)^{(m)} \circ q(\hat{U}^k) \equiv h\left( \underbrace{g(\cdot, r(p)) \circ ... \circ g(\cdot, r(p))}_{\text{composing } m \text{ times}} \left( q(\hat{U}^k) \right) \right). \tag{4}$$

Compared to autoregressive rollout in input space (Eq. 3), LE-PDE can significantly improve efficiency with a much smaller dimension of $\mathbf{z}^k \in \mathbb{R}^{d_z}$ compared to $U^k \in \mathbb{U}$. Here we do not limit the architecture for encoder, decoder and latent evolution models. Depending on the input $U^k$, the encoder $q$ and decoder $h$ can be a CNN or GNN with a (required) MLP head. In this work, we focus on input that is discretized as grid, so the encoder and decoder are both CNN+MLP, and leave other architecture (*e.g.* GNN+MLP) for future work. For static encoder $r$, it can be a simple MLP if the system parameter $p$ is a vector (*e.g.* equation parameters) or CNN+MLP if $p$ is a 2D or 3D tensor (*e.g.* boundary mask, spatially varying diffusion coefficient). We model the latent evolution model $g$

as an MLP with residual connection from input to output. The architectures used in our experiments, are detailed in Appendix C, together with discussion of its current limitations.

## 3.2 Learning objective

Learning surrogate models that can faithfully roll out long-term is an important challenge. Given discretized inputs $\{U^k\}, k = 1, ...K + M$, we introduce the following objective to address it:

$$L = \frac{1}{K} \sum_{k=1}^{K} (L_{\text{multi-step}}^k + L_{\text{recons}}^k + L_{\text{consistency}}^k). \tag{5}$$

$$\text{where} \begin{cases} L_{\text{multi-step}}^k = \sum_{m=1}^{M} \alpha_m \ell(\hat{U}^{k+m}, U^{k+m}), \\ L_{\text{recons}}^k = \ell(h(q(U^k)), U^k) \\ L_{\text{consistency}}^k = \sum_{m=1}^{M} \frac{||g(\cdot, r(p))^{(m)} \circ q(U^k) - q(U^{k+m})||_2^2}{||q(U^{k+m})||_2^2} \end{cases} \tag{6}$$

Here $\ell$ is the loss function for individual predictions, which can typically be MSE or L2 loss. $\hat{U}^{k+m}$ is given in Eq. (4). $L_{\text{recons}}^k$ aims to reduce reconstruction loss. $L_{\text{multi-step}}^k$ performs latent multi-step evolution given in Eq. (4) and compare with the target $U^{k+m}$ in *input* space, up to time horizon $M$. $\alpha_m$ are weights for each time step, which we find that $(\alpha_1, \alpha_2, ...\alpha_M) = (1, 0.1, 0.1, ...0.1)$ works well. Besides encouraging better prediction in input space via $L_{\text{multi-step}}^k$, we also want a stable long-term rollout in latent space. This is because in inference time, we want to mainly perform autoregressive rollout in latent space, and decode to input space only when needed. Thus, we introduce a novel latent consistency loss $L_{\text{consistency}}^k$, which compares the $m$-step latent rollout $g(\cdot, r(p))^{(m)} \circ q(U^k)$ with the latent target $q(U^{k+m})$ in *latent* space. The denominator $||q(U^{k+m})||_2^2$ serves as normalization to prevent the trivial solution that the latent space collapses to a single point. Taken together, the three terms encourage a more accurate and consistent long-term evolution both in latent and input space. In Sec. 4.4 we will investigate the influence of $L_{\text{consistency}}^k$ and $L_{\text{multi-step}}^k$.

## 3.3 Accelerating inverse optimization

In addition to improved efficiency for forward simulation, LE-PDE also allows more efficient inverse optimization, via backpropagation through time (BPTT) in latent space. Given a specified objective $L_d[p] = \sum_{k=k_s}^{k_e} \ell(U^k)$ which is a discretized version of $L_d[\mathbf{a}, \partial \mathbb{X}]$ in Sec. 2, we define the objective:

$$L_d[p] = \sum_{m=k_s}^{k_e} \ell_d(\hat{U}^m(p)) \tag{7}$$

where $\hat{U}^m = \hat{U}^m(p)$ is given by Eq. (4) setting $k = 0$ using our learned LE-PDE, which starts at initial state of $U^0$, encode it and $p$ into latent space, evolves the dynamics in latent space and decode to $\hat{U}^m$ as needed. The static latent embedding $\mathbf{z}_p = r(p)$ influences the latent evolution at each time step via $g(\cdot, r(p))$. An approximately optimal parameter $p$ can then be found by computing gradients $\frac{\partial L_d[p]}{\partial p}$, using optimizers such as Adam [61] (The gradient flow is visualized as the red arrows in Fig. 1). When $p$ is a boundary parameter, *e.g.* location of the boundary segments or obstacles, there is a challenge. Specifically, for CNN encoder $q$, the boundary information is typically provided as a binary mask indicating which cells are outside the simulation domain $\Omega$. The discreteness of the mask prevents the backpropagation of the model. Moreover, the boundary cells may interact sparsely with the bulk, which can lead to vanishing gradient during inverse optimization. To address this, we introduce a function that maps $p$ to a soft boundary mask with temperature, and during inverse optimization, anneal the temperature from high to low. This allows the gradient to pass through mask to $p$, and stronger gradient signal. For more information, see Appendix B.

## 4 Experiments

In the experiments, we aim to answer the following questions: (1) Does LE-PDE able to learn accurately the long-term evolution of challenging systems, and compare competitively with state-of-the-art methods? (2) How much can LE-PDE reduce representation dimension and improving speed, especially with larger systems? (3) Can LE-PDE improve and speed up inverse optimization? For the first and second question, since in general there is a fundamental tradeoff between compression

Table 1: Performance of models in 1D for scenarios **E1**,**E2**, **E3**. Accumulated error $= \frac{1}{n_x}\sum_{t,x}$ MSE. Representation dimension ($= S \times n_x$ here) is the number of dimensions to update at each time step. The bold values represent the best performance for experiments and underline shows second best.

| | | Accumulated Error ↓ | | | | | Runtime [ms] ↓ | | | | Representation dim ↓ | |
|---|---|---|---|---|---|---|---|---|---|---|---|---|
| | $(n_t, n_x)$ | WENO5 | FNO-RNN | FNO-PF | MP-PDE | LE-PDE (ours) | WENO5 | MP-PDE | LE-PDE full (ours) | LE-PDE evo (ours) | MP-PDE | LE-PDE (ours) |
| **E1** | (250, 100) | 2.02 | 11.93 | **0.54** | 1.55 | 1.13 | $1.9 \times 10^3$ | 90 | 20 | **8** | 2500 | **128** |
| **E1** | (250, 50) | 6.23 | 29.98 | **0.51** | 1.67 | 1.20 | $1.8 \times 10^3$ | 80 | 20 | **8** | 1250 | **128** |
| **E1** | (250, 40) | 9.63 | 10.44 | **0.57** | 1.47 | 1.17 | $1.7 \times 10^3$ | 80 | 20 | **8** | 1000 | **128** |
| **E2** | (250, 100) | 1.19 | 17.09 | 2.53 | 1.58 | **0.77** | $1.9 \times 10^3$ | 90 | 20 | **8** | 2500 | **128** |
| **E2** | (250, 50) | 5.35 | 3.57 | 2.27 | 1.63 | **1.13** | $1.8 \times 10^3$ | 90 | 20 | **8** | 1250 | **128** |
| **E2** | (250, 40) | 8.05 | 3.26 | 2.38 | 1.45 | **1.03** | $1.7 \times 10^3$ | 80 | 20 | **8** | 1000 | **128** |
| **E3** | (250, 100) | 4.71 | 10.16 | 5.69 | 4.26 | **3.39** | $4.8 \times 10^3$ | 90 | 19 | **6** | 2500 | **64** |
| **E3** | (250, 50) | 11.71 | 14.49 | 5.39 | **3.74** | 3.82 | $4.5 \times 10^3$ | 90 | 19 | **6** | 1250 | **64** |
| **E3** | (250, 40) | 15.94 | 20.90 | 5.98 | **3.70** | 3.78 | $4.4 \times 10^3$ | 90 | 20 | **8** | 1000 | **128** |

(reduction of dimensions to represent a state) and accuracy [62, 63], *i.e.* the larger the compression to improve speed, the more lossy the representation is, we will need to sacrifice certain amount of accuracy. Therefore, the goal of LE-PDE is to maintain a reasonable or competitive accuracy (maybe slightly underperform state-of-the-art), while achieving significant compression and speed up. Thus, to answer these two questions, we test LE-PDE in standard benchmarks of a 1D family of nonlinear PDEs to test its generalization to new system parameters (Sec. 4.1), and a 2D Navier-Stokes flow up to turbulent phase (Sec. 4.2). The PDEs in the above scenarios have wide and important application in science and engineering. In each domain, we compare LE-PDE's long-term evolution performance, speed and representation dimension with state-of-the-art deep learning-based surrogate models in the domain. Then we answer question (3) in Section 4.3. Finally, in Section 4.4, we investigate the impact of different components of LE-PDE and important hyperparameters.

## 4.1 1D family of nonlinear PDEs

**Data and Experiments**. In this section, we test LE-PDE's ability to generalize to unseen equations with different parameters in a given family. We use the 1D benchmark in [7], whose PDEs are

$$\left[\partial_t u + \partial_x(\alpha u^2 - \beta\partial_x u + \gamma\partial_{xx}u)\right](t,x) = \delta(t,x) \tag{8}$$

$$u(0,x) = \delta(0,x), \quad \delta(t,x) = \sum_{j=1}^{J} A_j\sin(\omega_j t + 2\pi\ell_j x/L + \phi_j) \tag{9}$$

Here the parameter $p = (\alpha, \beta, \gamma)$. The term $\delta$ is a forcing term [64] with $J = 5, L = 16$ and coefficients $A_j$ and $\omega_j$ sampled uniformly from $A_j \sim U[-0.5, 0.5]$, $\omega_j \sim U[-0.4, 0.4]$, $\ell_j \in \{1, 2, 3\}$, $\phi_j \sim U[0, 2\pi]$. Space is uniformly discretized to $n_x = 200$ in $[0, 16)$ and time is uniformly discretized to $n_t = 250$ points in $[0, 4]$. Space and time are further downsampled to resolutions of $(n_t, n_x) \in \{(250, 100), (250, 50), (250, 40)\}$. The $\partial_x(\alpha u^2)$ advection term makes the PDE nonlinear. There are 3 scenarios with increasing difficulty: **E1:** Burgers' equation without diffusion $p = (1, 0, 0)$; **E2**: Burgers' equation with variable diffusion $p = (1, \eta, 0)$ where $\eta \in [0, 0.2]$; **E3**: mixed scenario with $p = (\alpha, \beta, \gamma)$ where $\alpha \in [0, 3], \beta \in [0, 0.4]$ and $\gamma \in [0, 1]$. **E1** tests the model's ability to generalize to new conditions with same equation. **E2** and **E3** test the model's ability to generalize to novel parameters of PDE with the same family. We compare LE-PDE with state-of-the-art deep learning-based surrogate models for this dataset, specifically MP-PDE [7] (a GNN-based model) and Fourier Neural Operators (FNO) [14]. For FNO, we compare with two versions: FNO-RNN is the autoregressive version in Section 5.3 of their paper, trained with autoregressive rollout; FNO-PF is FNO improved with the temporal bundling and push-forward trick as implemented in [7]. To ensure a fair comparison, our LE-PDE use temporal bundling of $S = 25$ time steps as in MP-PDE and FNO-PF. We perform hyperparameter search over latent dimension of $\{64, 128\}$ and use the model with best validation performance. In addition, we compare with downsampled ground-truth (WENO5), which uses a classical 5th-order WENO scheme [65] and explicit Runge-Kutta 4 solver [66, 67] to generate the ground-truth data and downsampled to the specified resolution. For all models, we autoregressively roll out to predict the states starting at step 50 until step 250, and record the accumulated MSE, runtime and representation dimension (the dimension of state to update at each time step). Details of the experiments are given in Appendix D.

**Results**. The result is shown in Table 1. We see that since LE-PDE uses 7.8 to 39-fold smaller representation dimension, it achieves significant smaller runtime compared to the MP-PDE model

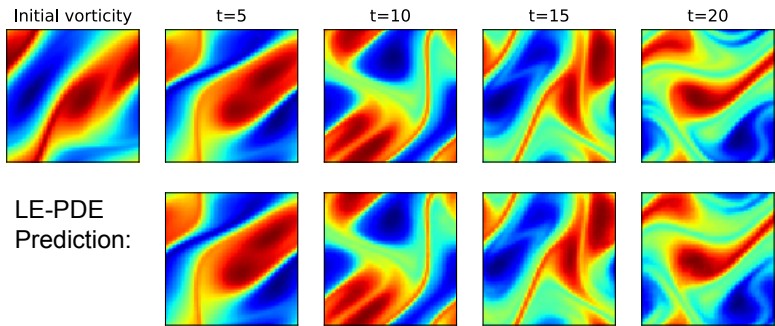

Figure 2: Visualization of rollout for 2D Navier-Stokes PDE ($Re = 10^4$), for ground-truth (upper) and LE-PDE (lower, trained with $\nu = 10^{-4}, N = 10^4$). LE-PDE captures detailed dynamics faithfully.

(which is much faster than the classical WENO5 scheme). Here we record the latent evolution time (LE-PDE evo) which is the total time for 200-step latent evolution, and the full time (LE-PDE full), which also includes decoding to the input space at each time step. The time for "LE-PDE evo" is relevant when the downstream application is only concerned with state at long-term future (*e.g.* [4]); the time for "LE-PDE full" is relevant when every intermediate prediction is also important. LE-PDE achieves up to 15× speed-up with "LE-PDE evo" and 4× speed-up with "LE-PDE full".

With above 7.8× compression and above 4× speed-up, LE-PDE still achieves competitive accuracy. For **E1** scenario, it significantly outperforms both original versions of FNO-RNN and MP-PDE, and only worse than the improved version of FNO-PF. For **E3**, LE-PDE outperforms both versions of FNO-RNN and FNO-PF, and the performance is on par with MP-PDE and sometimes better. For **E2**, LE-PDE outperforms all state-of-the-art models by a large margin. Fig. 4 in Appendix D shows our model's representative rollout compared to ground-truth. We see that during long-rollout, our model captures the shock formation faithfully. This 1D benchmark shows that LE-PDE is able to achieve significant speed-up, generalize to novel PDE parameters and achieve competitive long-term rollout.

### 4.2 2D Navier-Stokes flow

**Data and Experiments**. We test LE-PDE in a 2D benchmark [14] of Navier-Stokes equation. Navier-Stokes equation has wide application science and engineering, including weather forecasting, jet engine design, etc. It becomes more challenging to simulate when entering the turbulent phase, which shows multiscale dynamics and chaotic behavior. Specifically, we test our model in a viscous, incompressible fluid in vorticity form in a unit torus:

$$\partial_t w(t,x) + u(t,x) \cdot \nabla w(t,x) = \nu \Delta w(t,x) + f(x), \quad x \in (0,1)^2, t \in (0,T] \tag{10}$$

$$\nabla \cdot u(t,x) = 0, \qquad x \in (0,1)^2, t \in [0,T] \tag{11}$$

$$w(0,x) = w_0(x), \qquad x \in (0,1)^2 \tag{12}$$

Here $w(t,x) = \nabla \times u(t,x)$ is the vorticity, $\nu \in \mathbb{R}_+$ is the viscosity coefficient. The domain is discretized into $64 \times 64$ grid. We test with viscosities of $\nu = 10^{-3}, 10^{-4}, 10^{-5}$. The fluid is turbulent for $\nu = 10^{-4}, 10^{-5}$ ($Re \geq 10^4$). We compare state-of-the-art learning-based model Fourier Neural Operator (FNO) [14] for this problem, and strong baselines of TF-Net [26], U-Net [68] and ResNet [69]. For FNO, the FNO-2D performs autoregressive rollout, and FNO-3D directly maps the past 10 steps into all future steps. To ensure a fair comparison, here our LE-PDE uses past 10 steps to predict

Table 2: Performance of different models in 2D Navier-Stokes flow. Runtime is using the $\nu = 10^{-3}, N = 1000$ for predicting 40 steps in the future.

| Method | Representation dimensions | Runtime full [ms] | Runtime evo [ms] | $\nu = 10^{-3}$ $T = 50$ $N = 1000$ | $\nu = 10^{-4}$ $T = 30$ $N = 1000$ | $\nu = 10^{-4}$ $T = 30$ $N = 10000$ | $\nu = 10^{-5}$ $T = 20$ $N = 1000$ |
|---|---|---|---|---|---|---|---|
| FNO-3D [14] | 4096 | 24 | 24 | 0.0086 | 0.1918 | 0.0820 | 0.1893 |
| FNO-2D [14] | 4096 | 140 | 140 | 0.0128 | 0.1559 | 0.0834 | 0.1556 |
| U-Net [68] | 4096 | 813 | 813 | 0.0245 | 0.2051 | 0.1190 | 0.1982 |
| TF-Net [26] | 4096 | 428 | 428 | 0.0225 | 0.2253 | 0.1168 | 0.2268 |
| ResNet [69] | 4096 | 317 | 317 | 0.0701 | 0.2871 | 0.2311 | 0.2753 |
| **LE-PDE (ours)** | 256 | 48 | 15 | 0.0146 | 0.1936 | 0.1115 | 0.1862 |

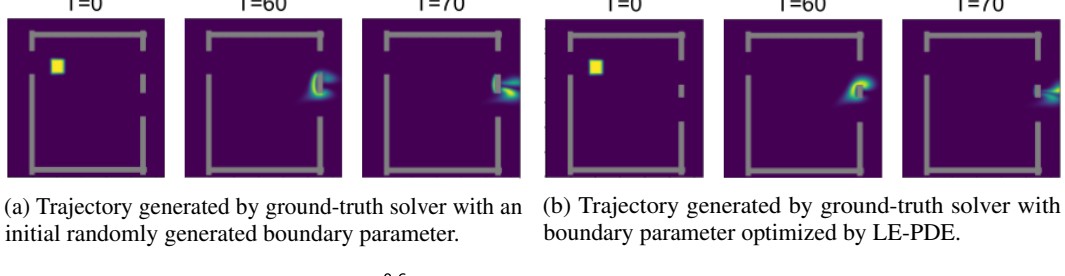

(a) Trajectory generated by ground-truth solver with an initial randomly generated boundary parameter.

(b) Trajectory generated by ground-truth solver with boundary parameter optimized by LE-PDE.

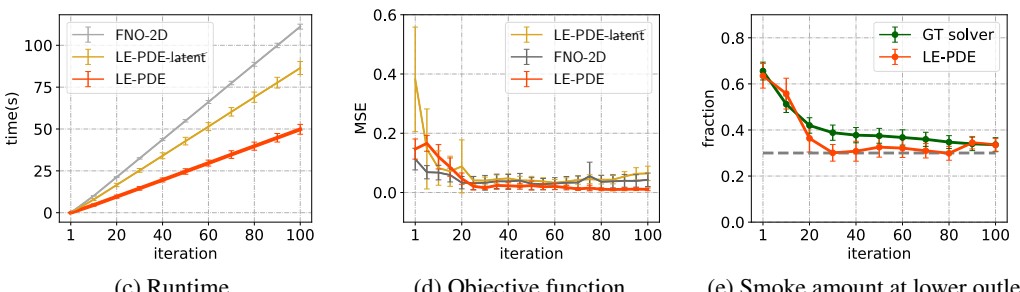

(c) Runtime.  (d) Objective function.  (e) Smoke amount at lower outlet.

Figure 3: Numerical results associated with inverse optimization of boundary (inlet and outlet designing) in Sec. 4.3. (a) shows a trajectory generated by ground-truth (GT) solver with an initial randomly generated boundary parameters (y-position of inlet and two outlets), with lower outlet passing 55.18% of smoke; (b) with optimized boundary parameters, with lower outlet passing 31.79% of smoke, very near the objective percentage of 30%. (c) Runtime and (d) learning curve (Eq. 7) for inverse optimization at different iteration steps; (e) For LE-PDE, fraction of smoke passing through the lower outlet computed by GT solver (green) and estimated by LE-PDE (orange). Error bar denotes 95% confidence interval over 50 runs with random initial conditions.

one future step and temporal bundling $S = 1$ (no bundling), the same setting as in FNO-2D. We use relative L2 norm (normalized by ground-truth's L2 norm) as metric, same as in [14].

**Results**. The results are shown in Table 2. Similar to 1D case, LE-PDE is able to compress the representation dimension by 16-fold. Hence, compared with FNO-2D which is also autoregressive, LE-PDE achieves 9.3-fold speed-up with latent evolution and 2.9-fold speed-up with full decoding. Compared with FNO-3D that directly maps all input time steps to all output times steps (which cannot generalize beyond the time range given), LE-PDE's runtime is still $1.6\times$ faster for latent evolution. For rollout L2 loss, LE-PDE significantly outperforms strong baselines of ResNet and U-Net, and TF-Net which is designed to model turbulent flow. Its performance is on par with FNO-3D with $\nu = 10^{-4}, N = 1000$ and the most difficult $\nu = 10^{-5}, N = 1000$ and slightly underperforms FNO-2D in other scenarios. Fig. 2 shows the visualization of LE-PDE comparing with ground-truth, under the turbulent $\nu = 10^{-4}, N = 10000$ scenario. We see that LE-PDE captures the detailed dynamics accurately. For more details, see Appendix E. To explore how LE-PDE can model and accelerate the simulation of systems with a larger scale, in Appendix F we explore modeling a 3D Navier-Stokes flow with millions of cells per time step, and show more significant speed-up.

### 4.3 Accelerating inverse optimization of boundary conditions

**Data and Experiments**. In this subsection, we set out to answer question (3), *i.e.* Can LE-PDE improve and speed up inverse optimization? We are interested in long time frame scenarios where the pre-defined objective $L_d$ in Eq. (7) depends on the prediction after long-term rollout. Such problems are challenging and have

Table 3: Comparison of LE-PDE with baselines.

|  | GT-solver Error (Model estimated Error) | Runtime $[s]$ |
|---|---|---|
| LE-PDE-latent | 0.305 (0.123) | 86.42 |
| FNO-2D | 0.124 (0.004) | 111.14 |
| **LE-PDE (ours)** | **0.035** (0.036) | **49.81** |

implications in engineering, *e.g.* fluid control [70, 71], laser design for laser-plasma interaction [4] and nuclear fusion [72]. To evaluate, we build a 2D Navier-Stokes flow in a family of boundary conditions using PhiFlow [73] as our ground-truth solver, shown in Fig. 3a, 3b. Specifically, we create a cubical boundary with one inlet and two outlets on a grid space of size $128^2$. We initialize the velocity and smoke on this domain and advect the dynamics by performing rollout. The objective

of the inverse design here is to optimize the boundary parameter $p$, *i.e.* the $y$-locations of the inlet and outlets, so that the amount of smoke passing through the two outlets coincides with pre-specified proportions $0.3$ and $0.7$. This setting is challenging since a slight change in boundary (up to a few cells) can have large influence in long-term rollout and the predefined objective.

As baseline methods, we use our LE-PDE's ablated version without latent evolution (essentially a CNN, which we call LE-PDE-~~latent~~) and the FNO-2D [14], both of which update the states in input space, while LE-PDE evolves in a 128-dimensional latent space ($128\times$ compression). To ensure a fair comparison, all models predict the next step using 1 past step without temporal bundling, and trained with 4-step rollout. We train all models with 400 generated trajectories of length 100 and test with 40 trajectories. After training, we perform inverse optimization w.r.t. the boundary parameter $p$ with the trained models using Eq. 7, starting with 50 initial configurations each with random initial location of smoke and random initial configuration of $p$. For LE-PDE-~~latent~~ and FNO-2D, they need to backpropagate through 80 steps of rollout in input space as in [45, 74], while LE-PDE backpropagates through 80 steps of latent rollout. Then the optimized boundary parameter is fed to the ground-truth solver for rollout and evaluate. For the optimized parameter, we measure the total amount of smoke simulated by the solver passing through two respective outlets and take their ratio. The evaluation metric is the average ratio across all 50 configurations: see also Appendix G.

**Results**. We observe that LE-PDE improves the overall speed by 73% compared with LE-PDE-~~latent~~ and by 123% compared with FNO-2D (Fig. 3c, Table 3). The result indicates a corollary of the use of low dimensional representation because Jacobian matrix of evolution operator is reduced to be of smaller size and suppresses the complexity associated with the chain rule to compute gradients of the objective function. While achieving the significant speed-up, the capability of the LE-PDE to design the boundary is also reasonable. Fig. 3d shows the loss of the objective function achieved the lowest value while the others are comparably large. The estimated proportion of smoke hit the target fraction $0.3$ at an early stage of design iterations and coincide with the fraction simulated by the ground-truth solver in the end (Fig. 3e). As Table 3 shows, FNO-2D achieves the lowest score in model estimated error from the target fraction $0.3$ while its ground-truth solver (GT-solver) error is $30\times$ larger. This shows "overfitting" of the boundary parameter by FNO-2D, *i.e.* the optimized parameter is not sufficiently generalized to work for a ground-truth solver. In this sense, LE-PDE achieved to design the most generalized boundary parameter: the difference between the two errors is the smallest among the others.

### 4.4 Ablation study

In this section, we investigate how each component of our LE-PDE influences the performance. Importantly, we are interested in how each of the three components: multi-step loss $L_{\text{multi-step}}$, latent consistency loss $L_{\text{consistency}}$ and reconstruction loss $L_{\text{recons}}$ contribute to the performance, and how the time horizon $M$ and the latent dimension $d_z$ influence the result. For dataset, we focus on representative scenarios in 1D (Sec. 4.1) and 2D (Sec. 4.2), specifically the **E2** scenario with $(n_t, n_x) = (250, 50)$ for 1D, and $(\nu = 10^{-5}, T = 20, N = 1000)$ scenario for 2D, which lies at mid- to difficult spectrum of each dataset. We have observed similar trends in other scenarios. From Table 4,

Table 4: Error for ablated versions of LE-PDE in 1D and 2D.

|  | 1D | 2D |
|---|---|---|
| **LE-PDE (ours)** | **1.127** | **0.1861** |
| no $L_{\text{multi-step}}$ | 3.337 | 0.2156 |
| no $L_{\text{consistency}}$ | 6.386 | 0.2316 |
| no $L_{\text{recons}}$ | 1.506 | 0.2025 |
| Time horizon $M = 1$ | 5.710 | 0.2860 |
| Time horizon $M = 3$ | 1.234 | 0.2010 |
| Time horizon $M = 4$ | 1.127 | 0.1861 |
| Time horizon $M = 6$ | 1.924 | 0.1923 |

we see that all three components $L_{\text{multi-step}}$, $L_{\text{consistency}}$ and $L_{\text{recons}}$ are necessary and pivotal in ensuring a good performance. The time horizon $M$ in the loss is also important. If too short (*e.g.* $M = 1$), it does not encourage accurate long-term rollout. Increasing $M$ helps reducing error, but will be countered by less number of examples (since having to leave room for more steps in the future). We find the sweet spot is at $M = 4$, which achieves a good tradeoff. In Fig. 6 in Appendix H, we show how the error and evolution runtime change with varying size of latent dimension $d_z$. We observe that reduction of runtime with decreasing latent dimension $d_z$, and that the error is lowest at $d_z = 64$ for 1D and $d_z = 256$ for 2D, suggesting the intrinsic dimension of each problem.

## 5 Discussion and Conclusion

In this work, we have introduced LE-PDE, a simple, fast and scalable method for accelerating simulation and inverse optimization of PDEs, including its simple architecture, objective and inverse optimization techniques. Compared with state-of-the-art deep learning-based surrogate models, we demonstrate that it achieves up to $128 \times$ reduction in the dimensions to update and up to $15 \times$ improvement in speed, while achieving competitive accuracy. Ablation study shows both multi-step objective and latent-consistency objectives are pivotal in ensuring accurate long-term rollout. We hope our method will make a useful step in accelerating simulation and inverse optimization of PDEs, pivotal in science and engineering.

## Acknowledgments and Disclosure of Funding

We thank Sophia Kivelson, Jacqueline Yau, Rex Ying, Paulo Alves, Frederico Fiuza, Jason Chou, Qingqing Zhao for discussions and for providing feedback on our manuscript. We also gratefully acknowledge the support of DARPA under Nos. HR00112190039 (TAMI), N660011924033 (MCS); ARO under Nos. W911NF-16-1-0342 (MURI), W911NF-16-1-0171 (DURIP); NSF under Nos. OAC-1835598 (CINES), OAC-1934578 (HDR), CCF-1918940 (Expeditions), NIH under No. 3U54HG010426-04S1 (HuBMAP), Stanford Data Science Initiative, Wu Tsai Neurosciences Institute, Amazon, Docomo, GSK, Hitachi, Intel, JPMorgan Chase, Juniper Networks, KDDI, NEC, and Toshiba.

The content is solely the responsibility of the authors and does not necessarily represent the official views of the funding entities.

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
