# Appendix

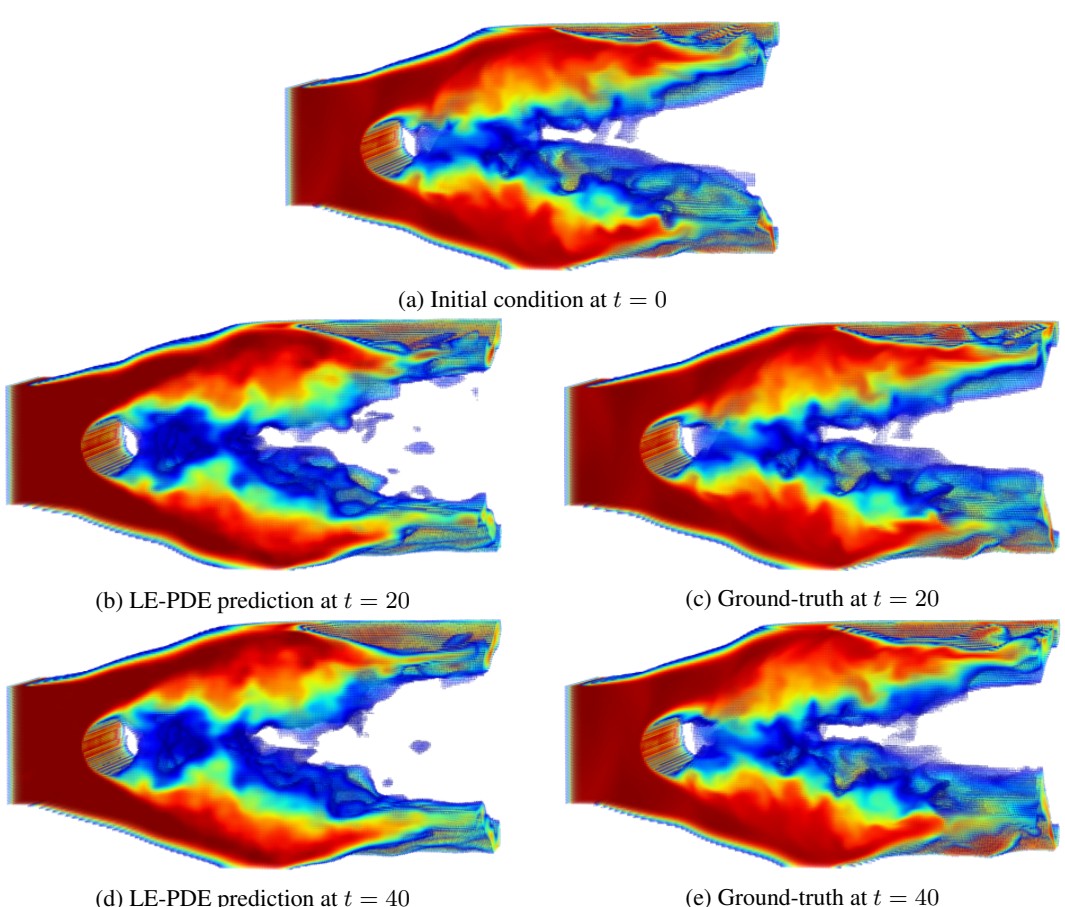

(a) Initial condition at $t = 0$

(b) LE-PDE prediction at $t = 20$

(c) Ground-truth at $t = 20$

(d) LE-PDE prediction at $t = 40$

(e) Ground-truth at $t = 40$

Figure S1: Visualization of LE-PDE testing on predicting the dynamics of turbulent 3D Navier-Stokes flow through the cylinder with a novel Reynolds number (detail in Appendix F). The input domain of size $2 \times 1 \times 1$ is discretized into a 3D grid of $256 \times 128 \times 128$, resulting in 4.19 million cells per time step. **Compression:** LE-PDE learns latent dynamics with latent dimension of $d_z = 128$, achieving a **130,000**$\times$ reduction in representation dimension, compared with 4.19 million cells times 4 features per cell ($\rho, v_x, v_y, v_z$) in input space. **Prediction quality:** The visualization is shown at a cross-section of $x = 50/128 \times 1$ along the direction of the cylinder. We see that compared with ground-truth (c)(e), LE-PDE (b)(d) captures the turbulent dynamics reasonably well, predicting both high-level and low-level dynamics in a qualitatively faithful way. This shows the scalability of LE-PDE to large-scale simulations of PDE. **Speed-up:** To predict the state at $t = 40$, on an Nvidia Quadro RTX 8000 48GB GPU, the ground-truth solver PhiFlow [73] uses 70.80s, an ablation our LE-PDE-latent without latent evolution (essentially a CNN) takes 1.03s, while our LE-PDE takes only 0.084s. LE-PDE achieves an **840**$\times$ speed-up compared to the ground-truth solver, and **12.3**$\times$ speed-up compared to the ablation model without latent evolution.

In the Appendix, we provide details that complement the main text. In Appendix A, we give a brief introduction to classical solvers. In Appendix B, we explain details about boundary interpolation and annealing technique used in Section 4.3. In Appendix C, we give full explanation on the architecture of LE-PDE used throughout the experiments. The following three sections explain details on parameter settings of experiments: 1D family of nonlinear PDEs (Appendix D), 2D Navier-Stokes flow (Appendix E) and 3D Navier-Stokes flow (Appendix F). In appendix G, we give details of boundary inverse optimization conducted in Section 4.3. In appendix H, we show ablation study for LE-PDE's important parameters. In Appendix I, we discuss the broader social impact of our work. In appendix J, we give comparison of trade-off between some metrics for LE-PDE and some

strong baselines. In addition, in Appendix K, we compare LE-PDE to another model exploiting latent evolution method from several aspects. We present the influence of varying noise amplitude with some tables in Appendix L. Finally, in Appendix M, we show the ablation study for various encoders in different scenarios.

## A Classical Numerical Solvers for PDEs

We refer the readers to [7] Section 2.2 and Appendix for a high-level introduction of the classical PDE solvers. One thing that is in common with the Finite Difference Method (FDM), Finite Volume Method (FVM) is that they all need to update the state of each cell at each time step. This stems from that the methods require discretization of the domain $\mathbb{X}$ and solution $\mathbf{u}$ into a grid $X$. For large-systems with millions or billions of cells, it will result in extremely slow simulation, as is also shown in Appendix F where a classical solver takes extremely long to evolve a 3D system with millions of cells per time step.

## B Boundary Interpolation and Annealing Technique

**Boundary interpolation**. In order to allow gradients to pass through to the boundary parameter $p$, we introduce a *continuous boundary mask* that continuously interpolates a discrete boundary mask and continuous variables. Here, for the later convenience, we regard a mask as a function from a grid structure $\mathbb{N}^{\times 2}_{\leq 128}$ to $[0, 1]$. Because boundary is composed by 1-dimensional segments, we use a 1-dimensional sigmoid function for the interpolation. Specifically, we define a sigmoid-interpolation function on a segment as a map to a real from a natural number $i$ conditioned by a pair of continuous variables $x_1, x_2$ and positive real $\beta$:

$$f(\,i \mid x_1, x_2, \beta) = \begin{cases} \text{sigmoid}(\frac{i-x_1}{\beta}), & i \leq x_1, \\ \text{sigmoid}(\frac{x_2-i}{\beta}), & x_2 \leq i, \\ \text{sigmoid}(\frac{GM_{-1}(|i-x_1|,|i-x_2|)}{\beta}), & x_1 < i < x_2. \end{cases} \tag{13}$$

Here, $x_1$ and $x_2$ are the location of the edge of the line-segment boundary, which is to be optimized during inverse optimization. $GM_{-1}(|i - x_1|, |i - x_2|) = (\frac{1}{2}(|i - x_1|^{-1} + |i - x_2|^{-1}))^{-1}$ denotes the harmonic mean[2], which is influenced more by the smaller of $|i - x_1|$ and $|i - x_2|$, so it is a *soft* version of the distance to the nearest edge inside the line segment of $x_1 < i < x_2$. When $\beta$ tends to 0, the function $f$ converges to a binary valued function: see also Fig. S2.

We define a continuous boundary function CB on a segment in a grid to be the pullback of the sigmoid-interpolation function with the projection to 1-dimensional discretized line (*i.e.*, take a projection of the pair of integers onto a 1-dimensional segment and apply $f$):

$$CB(\,(i,j) \mid (x_1, x_2), \beta) = \begin{cases} f(\,i \mid (x_1, x_2), \beta), & \text{if } (i, j) \text{ is in a horizontal segment,} \\ f(\,j \mid (x_1, x_2), \beta), & \text{if } (i, j) \text{ is in a vertical segment.} \end{cases} \tag{14}$$

Finally, a continuous boundary mask on a grid is obtained by (tranformation by a function $1 - x$ and) taking the maximum on a set of $CB$s on boundary segments on the grid (see also Fig. S3). The boundary interpolation allows the gradient to pass through the boundary mask and able to optimize the location of the edge of line segments (*e.g.* $x_1, x_2$).

**Boundary annealing**. As we see above, $\beta$ can be seen as a temperature hyperparameter, and the smaller it is, the more the boundary mask approximates a binary valued mask, and the less cells the boundary directly influences. At the beginning of the optimization, the parameter of the boundary (locations $x_1, x_2$ of each line segment) may be far away from the optimal location. Having a small temperature $\beta$ would result in vanishing gradient for the optimization, and very sparse interaction where the boundary mainly interact with its immediate neighbors, resulting in that very small gradient signal to optimize. Therefore, we introduce an annealing technique for the boundary optimization,

---

[2]$GM_\gamma(x, y) = (\frac{1}{2}(x^\gamma + y^\gamma))^{1/\gamma}$ is generalized mean with order $\gamma$. The harmonic mean $GM_{-1}(x, y)$ interpolates between arithmetic mean $GM_1(x, y) = \frac{1}{2}(x + y)$ and the minimum $GM_{-\infty}(x, y) = \min(x, y)$, and is influenced more by the smaller of $x$ and $y$.

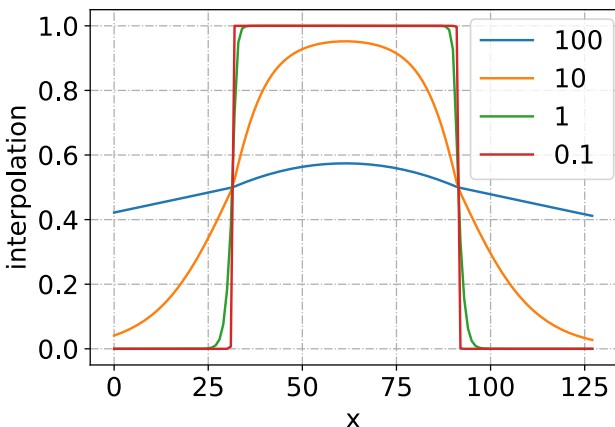

Figure S2: The interpolation of binary valued function by a sigmoid-interpolation function. Continuous variables $(x_1, x_2)$ are set to be $(31.5, 91.3)$. The continuous variables define edges of a continuous segment.

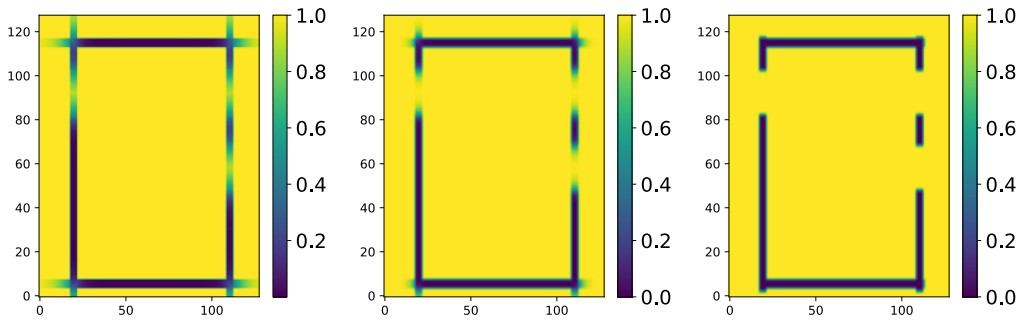

Figure S3: Continuous bounds with different parameters $\beta = 5$ (left), 2 (middle) and 0.01 (right). As $\beta$ decreases, the edges of the boundaries tend to have steeper slopes.

where at the beginning, we start at a larger $\beta_0$, and linearly tune it down until at the end reaching a much smaller $\beta$. The larger $\beta$ at the beginning allows denser gradient at the beginning of inverse optimization, where the location of the boundary can also influence more cells, producing more gradient signals. The smaller $\beta$ at the end allows more accurate boundary location optimization at the end, where we want to reduce the bias introduced by the boundary interpolation.

## C   Model Architecture for LE-PDE

Here we detail the architecture of LE-PDE, complementary to Sec. 3.1. This architecture is used throughout all experiment, with just a few hyperparameter (*e.g.* latent dimension $d_z$, number of convolution layers) depending on the dimension (1D, 2D, 3D) of the problem. We first detail the 4 architectural components of LE-PDE, and then discuss its current limitations.

**Dynamic encoder** $q$. The dynamic encoder $q$ consists of one CNN layer with (kernel-size, stride, padding) = $(3, 1, 1)$ and ELU activation, followed by $F_q$ convolution blocks, then followed by a flatten operation and an MLP with 1 layer and linear activation that outputs a $d_z$-dimensional vector $\mathbf{z}^k \in \mathbb{R}^{d_z}$ at time step $k$. Each of the $F_q$ convolution block consists of a convolution layer with (kernel-size, stride, padding) = $(4, 2, 1)$ followed by group normalization [74] (number of groups=2) and ELU activation [75]. The channel size of each convolution block follows the standard exponentially increasing pattern, *i.e.* the first convolution block has $C$ channels, the second has $C \times 2^1$ channels, ...

the $n^{\text{th}}$ convolution block has $C \times 2^{n-1}$ channels. The larger channel size partly compensates for smaller spatial dimensions of feature map for higher layers.

**Static encoder** $r$. For the static encoder $r$, depending on the static parameter $p$, it can be an $F_r$-layer MLP (as in 1D experiment Sec. 4.1 and 3D experiment Appendix F), or a similar CNN+MLP architecture as the dynamic encoder (as in Sec. 4.3 that takes as input the boundary mask). If using MLP, it uses $F_r$ layers with ELU activation and the last layer has linear activation. In our experiments, we select $F_r \in \{0, 1, 2\}$, and when $F_r = 0$, it means no layer and the static parameter is directly used as $\mathbf{z}_p$. The static encoder outputs a $d_{zp}$-dimensional vector $\mathbf{z}_p \in \mathbb{R}^{d_{zp}}$.

**Latent evolution model** $g$. The latent evolution model $g$ takes as input the concatenation of $\mathbf{z}^k$ and $\mathbf{z}_p$ (concatenated along the feature dimension), and outputs the prediction $\hat{\mathbf{z}}^{k+1}$. We model it as an MLP with residual connection from input to output, as an equivalent of the forward Euler's method in latent space:

$$\hat{\mathbf{z}}^{k+1} = \text{MLP}_g(\mathbf{z})^k + \mathbf{z}^k \tag{15}$$

In this work, we use the same $\text{MLP}_g$ architecture throughout all sections, where the $\text{MLP}_g$ consists of 5 layers, each layer has the same number $d_z$ of neurons as the dimension of $\mathbf{z}^k$. The first three layers has ELU activation, and the last two layers have linear activation. We use two layers of linear layer instead of one, to have an implicit rank-minimizing regularization [76], which we find performs better than 1 last linear layer.

**Decoder** $h$. Mirroring the encoder $q$, the decoder $h$ takes as input the $\mathbf{z}^{k+m} \in \mathbb{R}^{d_z}, m = 0, 1, ...M$, through an $\text{MLP}_h$ and a CNN with $F_h = F_q$ number of convolution-transpose blocks, and maps to the state $U^{k+m}$ at input space. The $\text{MLP}_h$ is a one layer MLP with linear activation. After it, the vector is reshaped into the shape of (batch-size, channel-size, *image-shape) for the $F_h$ convolution-transpose blocks. Then it is followed by a single convolution-transpose layer with (kernel-size, stride, padding)=$(3, 1, 1)$ and linear activation. Each convolution-transpose block consists of one convolution-transpose layer with (kernel-size, stride, padding) = $(4, 2, 1)$, followed by group normalization and an ELU activation. The number of channels also follows a mirroring of the encoder $q$, where the nearer to the output, the smaller the channel size with exponentially decreasing size.

**Limitations of current LE-PDE architecture**. The use of MLPs in the encoder and decoder has its benefits and downside. The benefit is that due to its flatten operation and MLP that maps to a much smaller vector $\mathbf{z}$, it can significantly improve speed, as demonstrated in the experiments in the paper. The limitation is that it requires that the training and test datasets to have the same discretization, otherwise a different discretization will result in a different flattened dimension making the MLP in the encoder and decoder invalid. We note that despite this limitation, it already encompasses a vast majority of applications where the training and test datasets share the same discretization (but with novel initial condition, static parameter $p$, etc.). Experiments in this paper show that our method is able to generalize to novel equations in the same family (Sec. 4.1), novel initial conditions (Sec. 4.2 and 4.3) and novel Reynolds numbers in 3D (Appendix F). Furthermore, our general 4-component architecture of dynamic encoder, static encoder, latent evolution model and decoder is very general and can allow future work to transcend this limitation. Future work may go beyond the limitation of discretization, by incorporating ideas from *e.g.* neural operators [34, 36], where the latent vector encodes the solution *function* $\mathbf{u}(\mathbf{x}, t)$ instead of the discretized states $U^k$, and the latent evolution model then models the latent dynamics of neural *operators* instead of functions.

Similar to a majority of other deep-learning based models for surrogate modeling (*e.g.* [13, 14]), the conservation laws present in the PDE is *encouraged* through the loss w.r.t. the ground-truth, but not generally *enforced*. Building domain-specific architectures that enforces certain conservation laws is out-of-scope of this work, since we aim to introduce a more general method for accelerating simulating and inverse optimizing PDEs, applicable to a wide scope of temporal PDEs. It is an exciting open problem, to build more structures into the latent evolution that obeys certain conservation laws or symmetries, potentially incorporating techniques *e.g.* in [77, 78]. Certain conservation laws can also be enforced in the decoder, for example similar to the zero-divergence as in [57].

# D    Details for experiments in 1D family of nonlinear PDEs

Here we provide more details for the experiment for Sec. 4.1. The details of the dataset have already been given in Section 4.1 and more detailed information can be found in [7] that introduced the benchmark.

**LE-PDE**. For LE-PDE in this section, the convolution and convolution-transpose layers are 1D convolutions, since the domain is 1D. We use temporal bundling steps $S = 25$, similar to the MP-PDE, so it based on the past $S = 25$ steps to predict the next $S = 25$ steps. The input has shape of (batch-size, $S$, $C_{in} = 1$, $n_x$), which[3] we flatten the $S$ and $C_{in}$ dimensions into a single dimension and feed the (batch-size, $S \times C_{in} = 25$, $n_x$) tensor to the encoder. For the convolution layers in encoder, we use starting channel size $C = 32$ and exponential increasing channels as detailed in Appendix C. We use $F_q = F_r = 4$ blocks of convolution (or convolution-transpose).

We perform search on hyperparameters of latent dimension $d_z \in \{64, 128\}$, loss function $\ell \in \{\text{MSE}, \text{RMSE}\}$, time horizon $M \in \{4, 5\}$, and number of layers for static encoder $F_r \in \{0, 1, 2\}$, and use the model with the best validation loss. We train for 50 epochs with Adam [61] optimizer with learning rate of $10^{-3}$ and cosine learning rate annealing [76] whose learning rate follows a cosine curve from $10^{-3}$ to 0.

**Baselines**. For baselines, we directly report the baselines of MP-PDE, FNO-RNN, FNO-PR and WENO5 as provided in [7]. Details for the baselines is summarized in Sec. 4.1 and more in [7].

**More explanation for Table 1**. The runtimes in Table 1 are for one full unrolling that predicts the future 200 steps starting at step 50, on a NVIDIA 2080 Ti RTX GPU. The "full" runtime includes the time for encoder, latent evolution, and decoding to all the intermediate time steps. The "evo" runtime only includes the runtime for the encoder and the latent evolution. The representation dimension, as explained in Sec. 4.1, is the number of feature dimensions to update at each time step. For baselines of MP-PDE, etc. it needs to update $n_x \times S \times 1$ dimensions, *i.e.* the consecutive $S = 25$ steps of the 1D space with $n_x$ cells (where each cell have one feature). For example, for $n_x = 100$, the representation dimension is $n_x \times S \times 1 = 100 \times 25 \times 1 = 2500$. In contrast, our LE-PDE uses a 64 or 128-dimensional latent vector to represent the same state, and only need to update it for every latent evolution.

**Visualization of LE-PDE rollout**. In Fig. 4, we show example rollout of our LE-PDE in the **E2** scenario and comparing with ground-truth. We see that LE-PDE captures the shock formation (around $x = 14$) faithfully, across all three spatial discretizations.

# E    Details for 2D Navier-Stokes flow

Here we detail the experiments we perform for Sec. 4.2. For the baselines, we use the results reported in [14]. For our LE-PDE, we follow the same architecture as detailed in Appendix C. Similar to other models (*e.g.* FNO-2d), we use temporal bundling of $S = 1$ (no bundling) and use the past 10 steps to predict one future step, and autoregressively rollout for $T - 10$ steps, then use the relative L2 loss over the all the predicted states as the evaluation metric. We perform search on hyperparameters of latent dimension $d_z \in \{128, 256\}$, loss function $\ell \in \{\text{MSE}, \text{RMSE}, \text{L2}\}$, time horizon $M \in \{4, T - 10\}$, number of epochs $\{200, 500\}$, and use the model with the best validation loss. The runtime in Table 2 is computed using an Nvidia Quadro RTX 8000 48GB GPU (since the FNO-3D exceeds the memory of the Nvidia 2080 Ti RTX 11GB GPU, to make a fair comparison, we use this larger-memory GPU for all models for runtime comparison).

# F    3D Navier-Stokes flow

To explore how LE-PDE can scale to larger scale turbulent dynamics and its potential speed-up, we train LE-PDE in a 3D Navier-Stokes flow through the cylinder using a similar 3D dataset in [12], generated by PhiFlow [73] as the ground-truth solver. The PDE is given by:

---

[3]Here $C_{in}$ is the number of input channels for $u(t, x)$. It is 1 since the $u(t, x)$ has only one feature.

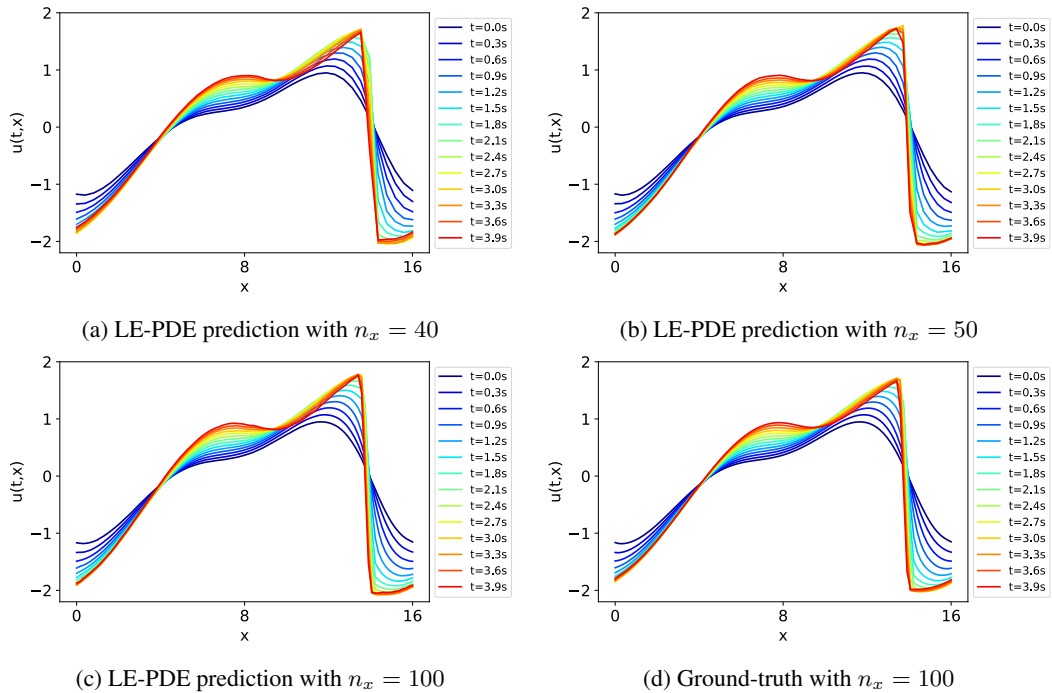

(a) LE-PDE prediction with $n_x = 40$

(b) LE-PDE prediction with $n_x = 50$

(c) LE-PDE prediction with $n_x = 100$

(d) Ground-truth with $n_x = 100$

Figure 4: Example rollout of LE-PDE for 200 steps (0 to 4s), with **E2** scenario that tests the models ability to generalize to new equations within the same family, for (a) $n_x = 40$, (b) $n_x = 50$, (c) $n_x = 100$, compared with ground-truth of (d) $n_x = 100$. The LE-PDE models in the plot are using the ones reported in Table 1. We see that LE-PDE captures the shock formation (around $x = 14$) very accurately and faithfully, across all three spatial discretizations.

$$\partial_t u_x + \mathbf{u} \cdot \nabla u_x = -\frac{1}{\rho}\nabla p + \nu\nabla \cdot \nabla u_x, \qquad (16)$$

$$\partial_t u_y + \mathbf{u} \cdot \nabla u_y = -\frac{1}{\rho}\nabla p + \nu\nabla \cdot \nabla u_y, \qquad (17)$$

$$\partial_t u_z + \mathbf{u} \cdot \nabla u_z = -\frac{1}{\rho}\nabla p + \nu\nabla \cdot \nabla u_z, \qquad (18)$$

$$\text{subject to } \nabla \cdot \mathbf{u} = 0. \qquad (19)$$

We discretize the space into a 3D grid of $256 \times 128 \times 128$, resulting in 4.19 million cells per time step. We generate 5 trajectories of length 500 with Reynolds number $\{55.5, 56.8, 58.0, 58.3, 58.6\}$ for training/validation set and test the model's performance on 2 additional trajectories with $\{57.4, 58.0\}$. All the trajectories have different initial conditions. We sub-sample the time every other step, so the time interval between consecutive time step for training is 2s. For LE-PDE, we follow the architecture in Appendix C, with $F_q = F_h = 5$ convolution (convolution-transpose) blocks in the encoder (decoder), latent dimension $d_z = 128$, and starting channel dimension of $C = 32$. We use time horizon $M = 4$ in the learning objective (Eq. 5), with $(\alpha_1, \alpha_2, \alpha_3, \alpha_4) = (1, 0.1, 0, 0.1)$ (we set the third step $\alpha_3 = 0$ due to the limitation in GPU memory). The Reynolds number $p = Re$ is copied 4 times and directly serve as the static latent parameter (number of layers $F_r$ for static encoder MLP $r$ is 0). This static encoder allows LE-PDE to generalize to novel Reynolds numbers. We use $\ell = $MSE. We randomly split 9:1 in the training/validation dataset of 5 trajectories, train for 30 epochs, save the model after each epoch, and use the model with the best validation loss for testing.

**Prediction quality**. In Fig. S1, we show the prediction of LE-PDE on the first test trajectory with a novel Reynolds number ($Re = 57.4$) and novel initial conditions. We see that LE-PDE captures the high-level and low-level turbulent dynamics in a qualitatively reasonable way, both at the tail and also

in the inner bulk. This shows the scalability of our LE-PDE to learn large-scale PDEs with intensive dynamics in a reasonably faithful way.

**Speed comparison**. We compare the runtime of our LE-PDE, an ablation LE-PDE-latent and the ground-truth solver PhiFlow, to predict the state at $t = 40$. The result is shown in Table 5. For the ablation LE-PDE-latent, its latent evolution model and the MLPs in the encoder and decoder are ablated, and it directly uses the other parts of encoder and decoder to predict the next step (essentially a 12-layer CNN). We see that our LE-PDE achieves a $70.80/0.084 \simeq 840\times$ speed up compared to the ground-truth solver on the same GPU. We see that w.r.t. LE-PDE-latent (a CNN) that is significantly faster than solver, our LE-PDE is still $1.03/0.084 = 12.3$ times faster. This shows that our LE-PDE can significantly accelerate the simulation of large-scale PDEs.

**Comparison of number of parameters**. We see that our LE-PDE uses much less number of parameters to evolve autoregressively than FNO. The most parameters of LE-PDE are mainly in the encoder and decoder, which is only applied once at the beginning and end of the evolution. Thus, LE-PDE achieves a much smaller runtime than FNO to evolve to t=40.

# G   Details for inverse optimization of boundary conditions

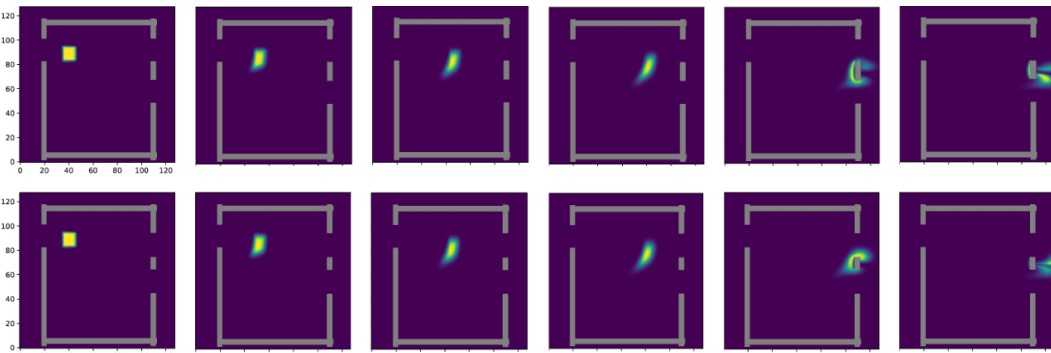

Figure S4: Trajectories generated by ground-truth solver with initial boundary parameter (upper) and optimized boundary parameter (lower).

**Objective function**. To define the objective function, we create masks $(o_1, o_2)$ that correspond to respective outlets of given a boundary. The masks are defined to be ones on the outlets' voids (see also Fig. S5). With the masks, we define the objective function in Sec. 3.3 that can measure the amount of smoke passing through the outlets:

$$L_d[p] = \sum_{i=1}^{2} \text{MSE}(t_i, \frac{\sum_{m=k_s}^{k_e}\langle o_i, \hat{U}^m(p)\rangle}{K}).$$

Here, $(t_1, t_2) = (0.3, 0.7)$, $K = \sum_{j=1}^{2}\sum_{m=k_s}^{k_e}\langle o_j, \hat{U}^m(p)\rangle$ and $\langle x, y \rangle = x^{\text{T}}y$. We set $k_s = 50$, *i.e.*, we use smoke at scenes after 50 time steps to calculate the amount of the smoke.

**LE-PDE**. The encoder $q$ and decoder $h$ have $F_q = F_h = 4$ blocks of convolution (or convolution-transpose) followed by MLP, as specified in Appendix C. The time step of input is set to be 1.

Table 5: Comparison of LE-PDE with baseline on runtime and representation dimension, in the 3D Navier-Stokes flow. The runtime is to predict the state at $t = 40$.

|  | Runtime (s) | Representation dimension | Error at $t = 40$ | # Paramters | # Parameters for evolution model | Training time (min) per epoch | Memory usage (MiB) |
|---|---|---|---|---|---|---|---|
| PhiFlow (ground-truth solver) on CPU | 1802 | $16.76 \times 10^6$ | - | - | - | - | - |
| PhiFlow (ground-truth solver) on GPU | 70.80 | $16.76 \times 10^6$ | - | - | - | - | - |
| FNO (with 2-step loss) | 7.00 | $16.76 \times 10^6$ | **0.1695** | 3,281,864 | 3,281,864 | 102 | 25,147 |
| FNO (with 1-step loss) | 7.00 | $16.76 \times 10^6$ | 0.3215 | 3,281,864 | 3,281,864 | 58 | 24,891 |
| LE-PDE-latent | 1.03 | $16.76 \times 10^6$ | 0.1870 | 71,396,976 | 71,396,976 | 69 | 21,361 |
| **LE-PDE (ours)** | **0.084** | **128** | 0.1947 | 65,003,120 | 83,072 | 65 | 25,595 |

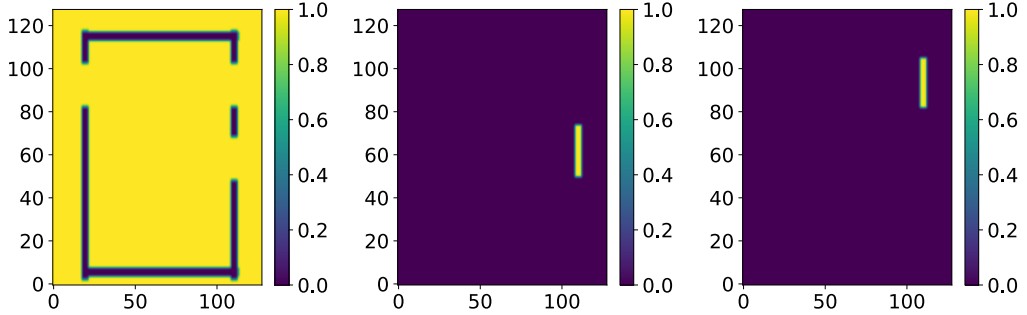

Figure S5: Figures of outlet masks for given a boundary mask. The left mask is a boundary mask, the middle mask $o_1$ corresponds to the lower outlet and the right $o_2$ the upper outlet.

The output of $q$ is a 128-dimensional vector $\mathbf{z}^k$. The latent evolution model $g$ takes as input the concatenation of $\mathbf{z}^k$ and 16-dimensional latent boundary representation $\mathbf{z}_p$ along the feature dimension, and outputs the prediction of $\hat{\mathbf{z}}^{k+1}$. Here, $\mathbf{z}_p$ is transformed by $r$ with the same layers as $q$, taking as input an boundary mask, where the boundary mask is a interpolated one specified in Appendix B. The architecture of the latent evolution model $g$ is the same as stated in Appendix C, with latent dimension $d_z = 128$.

**Parameters for inverse design**. We randomly choose 50 configurations for initial parameters. The sampling space is defined by the product of sets of inlet locations $\{79, 80, 81\}$, lower outlet locations $\{44, 45, 46, 47, 48, 49, 50\}$ and smoke position $\{0, 1\} \times \{-1, 0, 1\}$. We note that, even though we use the integers for the initial parameters, we can also use continuous values as initial parameters as long as the values are within the ranges of the integers. For one initial parameter, the number of the iterations of the inverse optimization is 100. During the iteration for each sampled parameter, we also perform linear annealing for $\beta$ of continuous boundary mask starting from 0.1 to 0.05. We also perform an ablation experiment with fixed $\beta = 0.05$ across the iteration. Fig. S6 shows the result. We see that without annealing, the GT-solver (ground-truth solver) computed Error (0.041) is larger than with annealing (0.035), and the gap estimated by the model and the GT-solver is much larger. This shows the benefit of using boundary annealing.

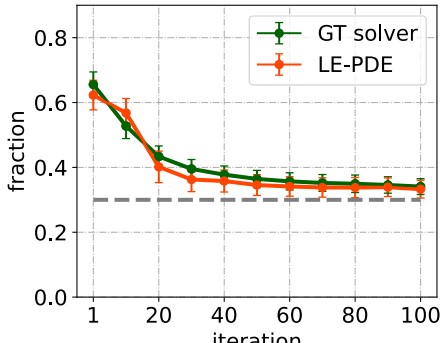

|  | GT-solver Error |
| :---: | :---: |
| **LE-PDE (ours)** | (Model estimated Error) |
| constant $\beta$ | 0.041 (0.032) |
| linear annealing $\beta$ | **0.035** (0.036) |

(b) Fractions estimated by ablated version of the inverse optimizer. Continuous boundary parameter $\beta$ in the ablated version is fixed across the iteration.

(a) Transition of fraction estimated by LE-PDE with fixed $\beta$. The difference (0.009) from fraction estimated by GT-solver is larger than that of LE-PDE with annealing (0.001) in Table 3.

Figure S6: Ablation study of annealer in the inverse design for continuous boundary parameter $\beta$.

**Model architecture of baselines**. We use the same notation used in Appendix C. LE-PDE-latent uses the dynamic encoder $q$ subsequently followed by the decoder $h$. Both $q$ and $h$ have the same number of layers $F_q = F_h = 4$. The output of $h$ is used as the input of the next time step. For the

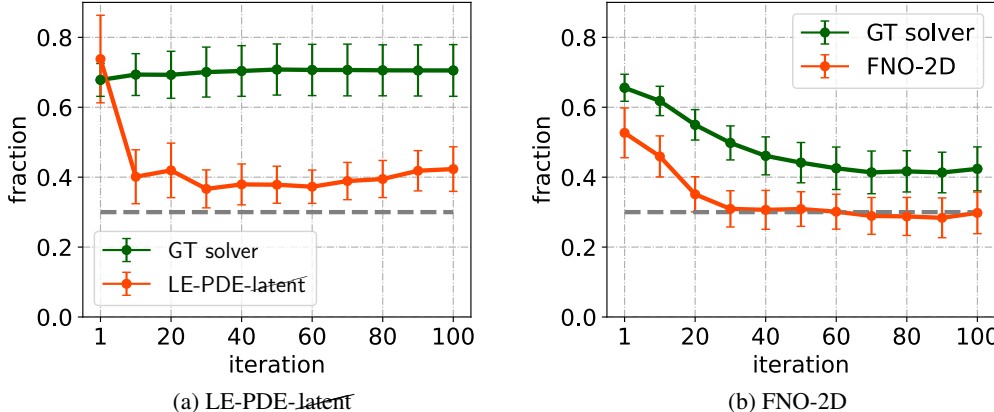

(a) LE-PDE-latent          (b) FNO-2D

Figure S7: Fraction of smoke passing through the lower outlet computed by GT solver and estimated by LE-PDE-latent and FNO-2D in Sec. 4.3. The dashed line denotes the objective of 0.3 fraction of smoke passing through the lower outlet.

FNO-2D model, we use the same architecture proposed in [14] with modes = 12 and width = 20. Fig. S7a and S7b are transition of fractions estimated by the ground-truth solver and the models with the boundary parameter under the inverse design. Compared with the one by our LE-PDE in Fig. 3e, we see that LE-PDE has much better GT-solver estimated fraction, and less gap between the fraction estimated by the GT-solver and the model.

# H    More ablation experiments with varying latent dimension

In this section, we provide complementary information to Sec. 4.4. Specifically, we provide tables and figures to study how the latent dimension $d_z$ influences the rollout error and runtime. Fig. 6 visualizes the results. Table 6 shows the results in the 1D **E2** $(n_t, n_x) = (250, 50)$ scenario that evaluate how LE-PDE is able to generalize to novel PDEs within the same family. And Table 7 shows the results in the 2D most difficult $(\nu = 10^{-5}, N = 1000)$ scenario.

**1D dataset**. From Table 6 and Fig. 6a, we see that when latent dimension $d_z$ is between 16 and 128, the accumulated MSE is near the optimal of $1 \sim 1.1$. It reaches minimum at $d_z = 64$. With larger latent dimensions, *e.g.* 256 or 512, the error slightly increases, likely due to the overfitting. With smaller latent dimension ($< 8$), the accumulated error grows significantly. This shows that the intrinsic dimension of this 1D problem with temporal bundling of $S = 25$ steps, is somewhere between 4 and 8. Below this intrinsic dimension, the model severely underfits, resulting in huge rollout error.

From the "runtime full" and "runtime evo" columns of Table 6 and also in Fig. 6b, we see that as the latent dimension $d_z$ decreases down from 512, the "runtime evo" has a slight decreasing trend down to 256, and then remains relatively flat. The "runtime full" also remains relatively flat. We don't see a significant decrease in runtime with decreasing $d_z$, likely due to that the runtime does not differ much in GPU with very small matrix multiplications.

**2D dataset**. From Table 7 and Fig. 6c, we see that similar to the 1D case, the error has a minimum in intermediate values of $d_z$. Specifically, as the latent dimension $d_z$ decreases from 512 to 4, the error first goes down and reaching a minimum of 0.1861 at $d_z = 128$. Then it slightly increase with decreasing $d_z$ until $d_z = 16$. When $d_z < 16$, the error goes up significantly. This shows that large latent dimension may results in overfitting, and the intrinsic dimension for this problem is somewhere between 8 and 16, below which the error will significantly go up. As the latent dimension decreases, the runtime have a very small amount of decreasing (from 512 to 256) but mostly remain at the same level. This relatively flat behavior is also likely due to that the runtime does not differ much in GPU with very small matrix multiplications.

Table 6: Performance evaluation for LE-PDE with different latent dimension on 1D dataset (E2-50 scenario). The accumulated error $= \frac{1}{n_x} \sum_{t,x} \text{MSE}$, summing over the predicted steps of 50-250, the same as in Table 1. The runtime is measured by rolling out with the same 200 steps, measured on a NVIDIA 2080 Ti RTX GPU, same as in Table 1. The default is with $d_z = 128$.)

| LE-PDE setting | cumulative error | runtime (full) (ms) | runtime (evolution) (ms) | # parameters | # parameters for latent evolution model |
|---|---|---|---|---|---|
| $d_z = 512$ | 2.778 | $16.3 \pm 2.6$ | $6.7 \pm 1.0$ | 4043648 | 1314816 |
| $d_z = 256$ | 2.186 | $15.0 \pm 0.8$ | $6.1 \pm 0.3$ | 2271360 | 329728 |
| $d_z = 128$ | 1.127 | $14.9 \pm 1.1$ | $6.0 \pm 0.4$ | 1630976 | 82944 |
| $d_z = 64$ | 0.994 | $14.4 \pm 1.0$ | $5.7 \pm 0.3$ | 1372224 | 20992 |
| $d_z = 32$ | 1.048 | $14.5 \pm 0.8$ | $5.8 \pm 0.4$ | 1258208 | 5376 |
| $d_z = 16$ | 1.041 | $14.1 \pm 0.9$ | $5.8 \pm 0.4$ | 1205040 | 1408 |
| $d_z = 8$ | 21.03 | $14.0 \pm 0.7$ | $5.6 \pm 0.2$ | 1179416 | 384 |
| $d_z = 4$ | 205.09 | $13.9 \pm 0.5$ | $5.7 \pm 0.3$ | 1166844 | 112 |

Table 7: Performance evaluation for LE-PDE with different latent dimension on 2D dataset ($\nu = 10^{-5}$ scenario. The Error is the relative L2 norm measured over 10 rollout steps, the same as in Table 2. The runtime is measured by rolling out with the same 10 steps, measured on a Nvidia Quadro RTX 8000 48GB GPU (same as in Table 2), and average over 100 runs (the number after $\pm$ is the std. of the 100 runs). The default is with $d_z = 128$.)

| LE-PDE setting | cumulative error | runtime (full) (ms) | runtime (evolution) (ms) | # parameters | # parameters for latent evolution model |
|---|---|---|---|---|---|
| $d_z = 512$ | 0.1930 | $16.2 \pm 1.1$ | $6.8 \pm 0.7$ | 6467184 | 1313280 |
| $d_z = 256$ | 0.1861 | $14.8 \pm 1.1$ | $5.8 \pm 0.4$ | 3384944 | 328960 |
| $d_z = 128$ | 0.2064 | $14.8 \pm 0.5$ | $5.9 \pm 0.4$ | 2089584 | 82560 |
| $d_z = 64$ | 0.2252 | $14.7 \pm 0.7$ | $6.0 \pm 0.7$ | 1503344 | 20800 |
| $d_z = 32$ | 0.2315 | $15.0 \pm 2.1$ | $5.9 \pm 0.5$ | 1225584 | 5280 |
| $d_z = 16$ | 0.2236 | $14.2 \pm 1.3$ | $5.8 \pm 0.6$ | 1090544 | 1360 |
| $d_z = 8$ | 0.3539 | $14.3 \pm 0.6$ | $5.7 \pm 0.3$ | 1023984 | 360 |
| $d_z = 4$ | 0.6353 | $14.2 \pm 0.5$ | $5.7 \pm 0.2$ | 990944 | 100 |

**More details in the ablation study experiments in Sec. 4.4.** For the ablation "Pretrain with $L_{\text{recons}}$", we pretrain the encoder and decoder with $L_{\text{recons}}$ for certain number of epochs, then freeze the encoder and decoder and train the latent evolution model and static encoder with $L_{\text{consistency}}$. Here the $L_{\text{multi-step}}$ is not valid since the encoder and decoder are already trained and frozen. For both 1D and 2D, we search hyperparameters of pretraining with $\{25, 50, 100\}$, and choose the model with the best validation performance.

# I  Broader social impact

Here we discuss the broader social impact of our work, including its potential positive and negative aspects, as recommended by the checklist. On the positive side, our work have huge potential implication in science and engineering, since many important problems in these domains are expressed as temporal PDEs, as discussed in the Introduction (Sec. 1). Although this work focus on evaluating our model in standard benchmarks, the experiments in Appendix F also show the scalability of our method to problems with millions of cells per time steps under turbulent dynamics. Our LE-PDE can be applied to accelerate the simulation and inverse optimization of the PDEs in science and engineering, *e.g.* weather forecasting, laser-plasma interaction, airplane design, etc., and may significantly accelerate such tasks.

We see no obvious negative social impact of our work. As long as it is applied to the science and engineering that is largely beneficial to society, our work will have beneficial effect.

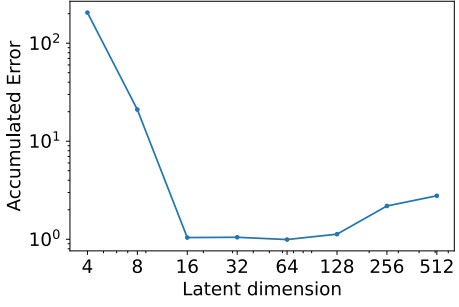

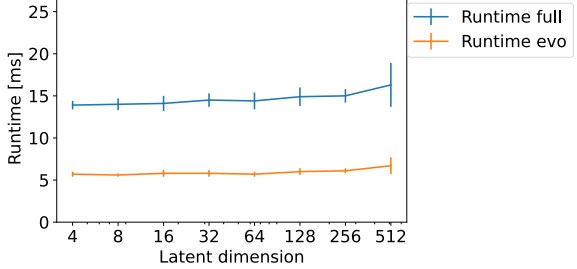

(a) Accumulated Error vs. latent dimension in 1D scenario. The Error is the relative L2 norm measured over 10 rollout steps, the same as in Table 2. The runtime is measured by rolling out with the same 10 steps, measured on a Nvidia Quadro RTX 8000 48GB GPU (same as in Table 2), and average over 100 runs (the number after $\pm$ is the std. of the 100 runs). The default is with $d_z = 128$.

(b) Runtime vs. latent dimension in 1D scenario

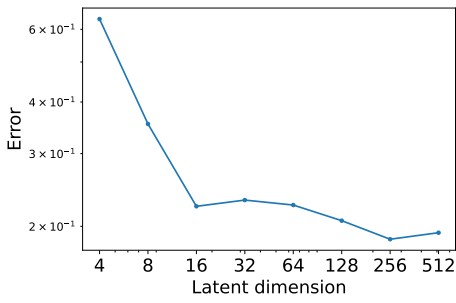

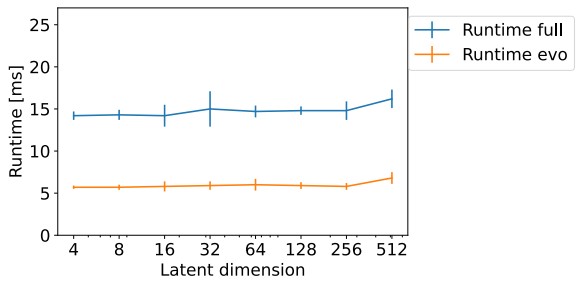

(c) Error vs. latent dimension in 2D scenario.

(d) Runtime vs. latent dimension in 2D scenario

Figure 6: Error vs. latent dimension $d_z$ for (a) 1D and (c) 2D scenario, and runtime vs. latent dimension $d_z$ for (b) 1D and (d) 2D scenario. We see that in 1D, the Error stays near optimum with latent size in [16, 128], and goes up outside the range. The runtime evo have a slight decreasing trend from latent dimension at 512 to 256, and stays relatively flat. For 2D, the Error decreases with increasing latent dimension, reaching an optimum at $d_z = 256$, and then slightly increases. Its runtime full have a slight decrease from latent dimension of 512 down to 256, and otherwise stays relatively flat.

## J   Pareto efficiency of FNO vs. LE-PDE

The following Table S8 shows the comparison of performance of FNO with varying hyperparameters. The hyperparameter search is performed on a 1D representative dataset E2-50. We evaluate the models (with varying hyperparameters) using the metric of the cumulative error and runtime. The most important hyperparameters for FNO are the "modes", which denotes the number of Fourier frequency modes, and "width", which denotes the channel size for the convolution layer in the FNO.

We also perform hyperparameter search on a 2D representative dataset with $\nu = 10^{-5}$. Table S9 shows the comparison of performance of FNO with varying hyperparameters. Hyperparameters to be varied and metrics for the evaluation are same as that of Table S8.

We can compare the above two tables with Table 6 and 7. We also create plots Figure S8 and S9 that compare the trade-off between several metrics shown in the tables for LE-PDE and FNO. Note that we provide both the total number of parameters (second last column) and number of parameters for latent evolution model (last column). The latter is also a good indicator since during long-term evolution, the latent evolution model is autoregressively applied while the encoder and decoder are only applied once. So the latent evolution model is the deciding component of the long-term evolution accuracy and runtime.

Table S8: Performance evaluation with FNO hyperparameter search on 1D dataset (E2-50 scenario.)

| FNO setting | cumulative error | runtime (full) (ms) | # parameters |
|---|---|---|---|
| modes=16, width=64 (default setting) | 2.379 | 21.2± 6.9 | 292249 |
| modes=16, width=128 | 3.107 | 21.7± 4.3 | 1138201 |
| modes=16, width=32 | 2.695 | 22.1± 7.4 | 78169 |
| modes=16, width=16 | 2.755 | 21.0± 5.7 | 23353 |
| modes=16, width=8 | 4.992 | 17.9± 1.2 | 9001 |
| modes=20, width=128 | 2.804 | 20.9± 1.1 | 1400345 |
| modes=20, width=64 | 2.626 | 19.3± 0.9 | 357785 |
| modes=12, width=64 | 2.899 | 19.6± 2.2 | 226713 |
| modes=8, width=64 | 2.240 | 19.7± 1.3 | 161177 |
| modes=4, width=64 | 2.326 | 19.2± 0.9 | 95641 |
| modes=8, width=32 | 2.366 | 18.2± 1.0 | 45401 |
| modes=8, width=16 | 2.505 | 18.1± 1.2 | 15161 |
| modes=8, width=8 | 5.817 | 18.4± 1.2 | 6953 |

Table S9: Performance evaluation with FNO hyperparameter search on 2D dataset ($\nu = 10^{-5}$ scenario.)

| FNO setting | L2 error | runtime (full)(ms) | # parameters |
|---|---|---|---|
| modes=12, width=20 (default setting) | 0.1745 | 42.7 ± 10.9 | 465717 |
| modes=12, width=40 | 0.1454 | 42.7 ± 4.2 | 1855977 |
| modes=12, width=10 | 0.2016 | 40.3 ± 5.4 | 117387 |
| modes=12, width=5 | 0.2398 | 45.5 ± 7.4 | 29922 |
| modes=16, width=20 | 0.1710 | 43.7 ± 4.2 | 824117 |
| modes=8, width=20 | 0.1770 | 43.1 ± 3.1 | 209717 |
| modes=4, width=20 | 0.1997 | 43.2 ± 4.8 | 56117 |
| modes=8, width=10 | 0.2109 | 42.2 ± 4.8 | 53387 |
| modes=8, width=5 | 0.2415 | 43.3 ± 4.3 | 13922 |

From the comparison, we see that:

- For 1D dataset, LE-PDE Pareto-dominates FNO in error vs. runtime plot (Fig. S8(a)). FNO's best cumulative error is 2.240, and runtime is above 17.9ms, over the full hyperparameters combinations (number of parameter varying from 6953 to 1.4M). In comparison, our LE-PDE achieves much better error and runtime over a wide parameter range: for $d_z$ from 16 to 64, LE-PDE's cumulative error $\leq 1.05$, runtime $\leq 14.5$ms, latent runtime $\leq 5.8$ms, (which uses 1408 to 82944 number of parameters for latent evolution model, and 1.2-1.4M total parameters). In terms of cumulative error vs. #parameter plot (Fig. S9(a)), the LE-PDE with evolution model typically has less parameters than FNO, which in turn also have less parameters than LE-PDE with full model. This makes sense, as the latent evolution requires much less parameters. Adding the encoder and decoder, LE-PDE may have more #parameters. But still it is the evolution parameter that is the most important for long-term evolution.

- For 2D dataset, FNO's cumulative error is slightly better than LE-PDE, but its runtime is significantly larger (Fig. S8(b)). Concretely, the best FNO achieves an error of 0.1454 while the best LE-PDE's error is 0.1861. FNO's runtime is above 40ms, while LE-PDE's runtime is generally below 15ms and latent evolution runtime is below 6ms. LE-PDE uses larger total number of parameters but much less number of parameters for latent evolution model. Also, similar to 1D, in terms of error vs. #parameter plot (Fig. S9(b)), the LE-PDE with evolution model typically has much less parameters than FNO, which in turn also typically have less parameters than LE-PDE with full model.

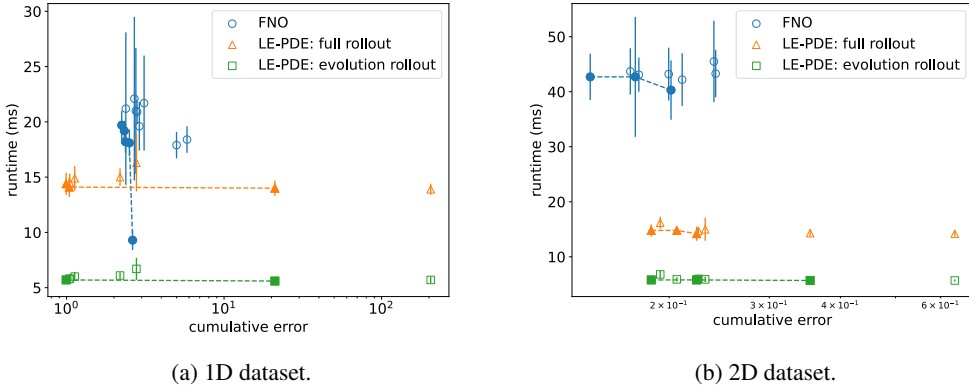

(a) 1D dataset.

(b) 2D dataset.

Figure S8: Comparison of trade-off between cumulative error and runtime of LE-PDE and FNO for 1D and 2D dataset. Dotted line connected to filled marker is Pareto frontier for respective model.

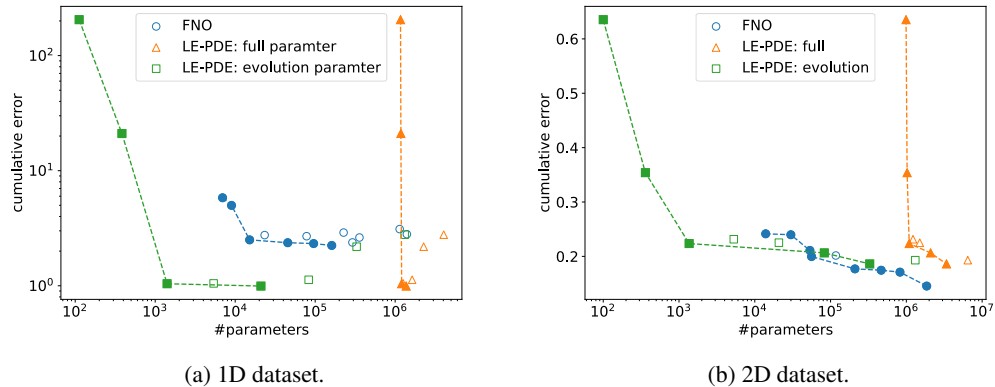

(a) 1D dataset.

(b) 2D dataset.

Figure S9: Comparison of trade-off between number of parameters and cumulative error of LE-PDE and FNO for 1D and 2D dataset. Dotted line connected to filled marker is Pareto frontier for respective model.

**Which family of PDEs can our LE-PDE apply:**. we can think of a PDE as a ground-truth model that evolves the *state* of a physical system. Typically, the states show more global, dominant features, and can be described by a state vector with much smaller dimension than the original discretization. Our LE-PDE exploit this *compressivity of state* to evolve the system in latent space and achieve speedup, and as long as the PDE does not significantly increase the spatial complexity of the state as it evolves (e.g. developing finer and finer spatial details as in 2-stream instability in of plasma [75]), our method can apply. Most of the PDEs satisfy the above requirements that the state are compressible and does not significantly increase its complexity, so our LE-PDE can be apply to most PDEs. Since any compression of state can incur a possible increase of error (possibly large or small, as the Pareto frontier of "error vs. runtime" and "error vs. #parameter" in Fig. S8 and S9 show for our LE-PDE and FNO), the more important/relevant question is then "what is the tradeoff of error vs. runtime we want for the given PDE", since we can design the encoder of LE-PDE with varying amount of compression. For example, we can design an encoder with minimal compression, so runtime reduction is low but can guarantee to retain low error, or with a much more aggressive compression (like in our 2D and 3D experiments), but can still achieve minimal increase of error. The amount of compression is a hyperparameter which can be obtained via validation set. Theoretically studying the best amount of compression that achieves a good tradeoff will be left for an exciting future work.

# K    Comparison of LE-PDE with LFM

To compare our LE-PDE with the Latent Field Model method (LFM) proposed in [18], we perform additional experiments in the representative 1D and 2D datasets in Section 4.4. We perform the ablation study where we (a) remove MLP in our model, (b) use LFM objective but maintain MLP, and (c) full LFM: remove MLP, use LFM objective, while all other aspects of training is kept the same. We use PyTorch's jvp function in autograd to compute the Jacobian-vector product and carefully make sure that our implementation is correct. Table S10 is the comparison table.

Table S10: Performance comparison of LE-PDE with LFM, for 1D dataset E2-50 scenario.

| LE-PDE setting | cumulative error | runtime (full) (ms) | runtime (evolution) (ms) | # parameters | # parameters for latent evolution model |
|---|---|---|---|---|---|
| LE-PDE (ours) | 1.127 | $14.9 \pm 1.1$ | $6.0 \pm 0.4$ | 1630976 | 82944 |
| (a) without MLP | 7.930 | $17.2 \pm 6.0$ | $8.3 \pm 0.4$ | 2730368 | 1580544 |
| (b) with LFM objective | 58.85 | $15.7 \pm 1.5$ | $6.5 \pm 0.6$ | 1630976 | 82944 |
| (c) full LFM: without MLP, with LFM objective | 26.12 | $15.7 \pm 1.3$ | $8.4 \pm 0.7$ | 2730368 | 1580544 |

Table S11 shows the comparison result of LE-PDE with LFM obtained by performing additional experiments on the representative 2D dataset in Section 4.4.

Table S11: Performance comparison of LE-PDE with LFM, for 2D dataset $\nu = 1e\text{-}5$ scenario.

| LE-PDE setting | cumulative error | runtime (full) (ms) | runtime (evolution) (ms) | # parameters | # parameters for evolution model |
|---|---|---|---|---|---|
| LE-PDE (ours) | 0.1861 | $14.8 \pm 1.1$ | $5.8 \pm 0.4$ | 3384944 | 328960 |
| (a) without MLP | 0.2120 | $16.6 \pm 2.2$ | $9.2 \pm 0.8$ | 2126960 | 1181184 |
| (b) with LFM objective | 0.4530 | $15.8 \pm 2.3$ | $6.2 \pm 0.6$ | 3384944 | 328960 |
| (c) full LFM: without MLP, with LFM objective | 0.6315 | $16.2 \pm 1.9$ | $9.1 \pm 0.4$ | 2126960 | 1181184 |

From the above tables, we see that without MLP, it actually results in worse performance (ablation (a)), and with LFM objective, the error is larger, likely due to that the dataset are quite chaotic and LFM may not adapt to the large time range in these datasets.

# L    Influence of varying noise amplitude

Here, we perform additional experiments on how the noise affects the performance, on the representative 1D used in Section 4.4 "Ablation Study". Table S12 shows the results. Specifically, we add random fixed Gaussian noise to the training, validation and test sets of the dataset, with varying amplitude. The noise is independently added to each feature of the dataset. It is also "fixed" in the sense that once added to the dataset, the noise is freezed and not re-sampled. This mimics the real world setting where random observation noise can corrupt the observation and we never have the ground-truth data to train and evaluate from.

We also perform experiments similar to Section L on a 2D representative dataset used in Section 4.4 "Ablation Study". Table S13 shows the results.

Note that the value range of both datasets are within $[-2, 2]$. From Table S12, we see that LE-PDE's cumulative error stays excellent ($\leq 1.456$) with noise amplitude $\leq 10^{-3}$, much smaller than state-of-the-art MP-PDE's error of $1.63$ and FNO-PF's $2.27$. Even with noise amplitude of $10^{-2}$, the LE-PDE's error of $2.612$ still remains reasonable.

From Table S13, we see that LE-PDE is quite resilient to noise, with error barely increases for noise amplitude up to $2 \times 10^{-2}$, and only shows minimal increase at noise level of $10^{-1}$. As a context, U-Net's error is $0.1982$ and TF-Net's error is $0.2268$ (Table 2 in main text).

In summary, in the 1D and 2D datasets, we see that LE-PDE shows good robustness to Gaussian noise, where the performance is reasonable where the ratio of noise amplitude to the value range can

Table S12: Evaluation of cumulative error of LE-PDE on 1D dataset (E2-50 scenario) with varying noise amplitude. The amplitude is the standard deviation of the diagonal Gaussian and the value range of the state $u(t, x)$ is within $[-2, 2]$.

| Noise amplitude | cumulative error |
|---|---|
| 0 (default) | 1.127 |
| $10^{-5}$ | 1.253 |
| $10^{-4}$ | 1.268 |
| $10^{-3}$ | 1.456 |
| $10^{-2}$ | 2.612 |
| $2 \times 10^{-2}$ | 4.102 |
| $5 \times 10^{-2}$ | 9.228 |

Table S13: Evaluation of cumulative error of LE-PDE on 2D dataset ($\nu = 10^{-5}$ scenario) with varying noise amplitude. The amplitude is the standard deviation of the diagonal Gaussian and the value range of the state $u(t, x)$ is within $[-2, 2]$.

| Noise amplitude | cumulative error |
|---|---|
| 0 (default) | 0.1861 |
| $10^{-5}$ | 0.1880 |
| $10^{-4}$ | 0.1862 |
| $10^{-3}$ | 0.1866 |
| $10^{-2}$ | 0.1897 |
| $2 \times 10^{-2}$ | 0.1875 |
| $5 \times 10^{-2}$ | 0.1910 |
| $10^{-1}$ | 0.2012 |

go up to 0.25% in 1D and 2.5% in 2D. The smaller robustness in the 1D Burgers' dataset may be due to that it is a 200-step rollout and the noise may make the model uncertain about the onset of shock formation.

## M   Ablation of LE-PDE using pretrained autoencoder or VAE

In addition, we perform two ablation experiments that explore performing data reduction first and then learn the evolution in latent space: (a) pretrain an autoencoder with states from all time steps, then freeze the autoencoder and train the latent evolution model. This mimics the method in [79]. (b) the encoder and decoder of LE-PDE is replaced with a VAE, first pre-trained with ELBO on all time steps, then freeze the encoder and decoder and train the latent evolution model. All other aspects of the model architecture and training remains the same. The result is shown in the Table S14 and Table S15 for the 1D and 2D datasets in Section 4.4 of "Ablation study".

Table S14: Ablation study of LE-PDE using pretrained autoencoder or VAE, for 1D dataset (E2-50 scenario.)

| LE-PDE setting | Cumulative error |
|---|---|
| LE-PDE (ours) | 1.127 |
| (a) pretrain autoencoder | 1.952 |
| (b) pretrained VAE | 1.980 |

From Table S14 and S15, we see that performing pre-training results in a much worse performance, since the data reduction only focuses on reconstruction, without consideration for *which* latent state is best used for evolving long-term into the future. On the other hand, our LE-PDE trains the components jointly with a novel objective that not only encourages better reconstruction, but also long-term evolution accuracy both in latent and input space. We also see that VAE as data-reduction performs worse than autoencoder, since the dynamics of the system is deterministic, and having a stochasticity from the VAE does not help.

Table S15: Ablation study of LE-PDE using pretrained autoencoder or VAE, for 2D dataset ($\nu = 10^{-5}$ scenario.)

| LE-PDE setting | Cumulative error |
|---|---|
| LE-PDE (ours) | 0.1861 |
| (a) pretrain autoencoder | 0.2105 |
| (b) pretrained VAE | 0.2329 |