# OpenReview forum: "Learning to Accelerate Partial Differential Equations via Latent Global Evolution"
_NeurIPS.cc/2022/Conference — NeurIPS 2022 Accept_

### Official Review · Reviewer_bK7j · 2022-06-21

**Rating:** 7
**Confidence:** 3
**Soundness:** 4 excellent
**Presentation:** 3 good
**Contribution:** 4 excellent

**Summary:**

The paper proposes a neural PDE solver based on learning how to represent the system with an evolving latent global state.
The advantage is reducing the cost of approximating PDE solutions without significantly reducing accuracy.
The method can also be used to solve some inverse problems.


**Questions:**

## Major

* In the objective defined in line 127, is the right hand side missing integration over $x$? If not, then there is a different objective for every $x$ location. How are these combined? (Also, less importantly, what does the subscript $d$ refer to?)

* In Table 1, why is the WEBO5 accumulated error so large if this is the ground truth method?

## Minor

* As someone not familiar with this field,
I found the subsection on "Deep learning-based surrogate modeling" reviewed autoregressive methods very clearly.
However the 1-sentence summary of "neural operators" was too brief for me to understand.
Can you expand on the details of "neural networks that approximate a mapping between infinite-dimensional functions",
as this seems non-trivial to do.
* Line 107. In what sense is $f$ a partial function?
* Table 1. Why is runtime for FNO not included?
* Line 233-234. The text says you start with $n_t=200$ then downsample. But downsampling is to $n_t=250$, which is a larger value!
Is there a typo here? If not, why not keep the original value of $n_t$?
* Line 314: "our LE-PDE's ablated version without latent evolution". Can you describe this in more detail?
I couldn't find a description in the main paper.
(Maybe it's somewhere in the supplement - in which case adding a reference would be helpful.)
* Line 354 "Increasing M... will be countered by less number of examples (since having to leave room for more steps in the future)"
I don't understand this, can you explain in more detail?

**Limitations:**

The paper covers technical and ethical limitations clearly in the supplementary material.
I agree with the authors' assessment that there are no obvious negative social impacts.
I also appreciate that the authors included experimental results where existing methods (slightly) outperform the proposed approach.
This helps readers get a clear picture of the relative strengths of the methods in different scenarios.

**Strengths And Weaknesses:**

## Summary (and strengths)

* Originality. The paper proposes an approach which is simple to understand and very general (allowing any architecture to be used for the various components e.g. encoders, decoders). Furthermore, it appears novel to me, although I'm not familiar with the neural PDEs literature.
* Quality. The proposed method is always at least competitive with existing methods, and outperforms them in some important examples. The experiments are thorough, covering all the important ablations I could think of while reading the paper.
* Clarity. Overall the paper is clearly written and easy to understand, but there are some minor problems discussed below.
* Significance.  The authors make the case for the importance of neural PDE solver well.

In my opinion this is a good paper and clearly above the acceptance threshold.

## Weaknesses

I found few important weaknesses in the paper. Some minor points are listed below and in the "Questions" section.

While the paper is generally well written, there are a few typos and confusingly worded sentences.
Most of these are unimportant, but below I list a few which made the paper harder to understand:

* Line 75 "flow probe" -> "flow to probe"?
* Line 89 "location... that satisfies given" Remove "that satisfies"?
* Line 122 "LE-PDE relieves of local evolution" I don't understand "relieves" in this context - use another word?
* Line 205 "The prevents the gradient to pass through to the boundary parameter $p$ such as continuous location." I'm struggling to parse this sentence, though I think the point is that it's not possible to backprop through discrete variables.
* Line 236 "$\eta \in [0,0,2]$" Should this say "$[0,2]$"?
* Line 324 "average amount of the advected smoke simulated by the solver" I don't follow what this means and I can't find the exact details in the supplement.

---

> ### Author Response · Authors · 2022-08-02
> **Response (2)**
>
> > Re: Table 1. Why is runtime for FNO not included?
>
> Answer: In Table 1, the results besides LE-PDE are provided by reference [1], which does not include the runtime for FNO. Since in 1D dataset, MP-PDE generally achieves the best performance, we compare mainly with MP-PDE.
>
> [1] Brandstetter, Johannes, Daniel Worrall, and Max Welling. "Message passing neural PDE solvers." ICLR 2022, arXiv preprint arXiv:2202.03376 (2022).
>
>
> > Re: Line 233-234. The text says you start with nt=200 then downsample. But downsampling is to nt=250, which is a larger value! Is there a typo here? If not, why not keep the original value of nt?
>
> Answer: thanks for catching the typo. $n_t$ should start with 250. We have updated it in the revised version.
>
>
> > Re: Line 314: "our LE-PDE's ablated version without latent evolution". Can you describe this in more detail? I couldn't find a description in the main paper. (Maybe it's somewhere in the supplement - in which case adding a reference would be helpful.)
>
> Answer: Thanks for asking. In the “LE-PDE's ablated version without latent evolution”, we maintain the encoder and decoder of LE-PDE and removing the latent evolution operator. To predict state $U^{t+1}$, we do $\hat{U}^{t+1}=decoder(encoder(U^t))$.
>
> > Re: Line 354 "Increasing M... will be countered by less number of examples (since having to leave room for more steps in the future)" I don't understand this, can you explain in more detail?
>
> Answer: Take the 1D dataset for example. Since the temporal bundling steps is 25 and total number of time step is 250, with multi-step M=1, the dataset consists of data with time steps of
>
> (input: [1,2,...25], target [26,27,...50]),
>
> (input: [2,3,...26], target [27,28,...51]),
>
> …
>
> (input: [201,202,...225], target [226,227,...250]),
>
> in total can have 201 number of data.
>
> If M=4, which means that the model need to rollout 4 steps and compare with target, the dataset would consist of data with time steps of
>
> (input: [1,2,...25], target [26,27,...50,51,...75,76,...100,101,....125]),
>
> (input: [2,3,...26], target [27,28,...51,52,...76,77,...101,102,....126]),
>
> …
>
> (input: [125,126,...150], target [151,...250])
>
> In total can have only 125 number of data.
>
> We see that since we need to have a larger length of target with larger M, it results in smaller number of data for each simulation.
>
> **Summary**
>
> Thanks again for the detailed comments and questions! We hope that we have answered your questions and resolved your concerns.

---

> ### Author Response · Authors · 2022-08-02
> **Response (1)**
>
> We thank the reviewer for positive review, and glad that the reviewer recognizes the originality, quality, clarity and significance of the work. Below we answer the questions raised by the reviewer.
>
> > Re: Line 75 "flow probe" -> "flow to probe"?
> Line 89 "location... that satisfies given" Remove "that satisfies"?
> Re: Line 122 "LE-PDE relieves of local evolution" I don't understand "relieves" in this context - use another word?
>
> Answer: Fixed in the revised script.
>
> > Re: Line 205 "The prevents the gradient to pass through to the boundary parameter p such as continuous location." I'm struggling to parse this sentence, though I think the point is that it's not possible to backprop through discrete variables.
>
> Answer: yes, your understanding is correct. Typically the boundary is passed in as a binary mask as input, so a discrete value where gradient cannot pass through. We have improved the sentence in the revised script.
>
> > Re:  "η∈[0,0,2]" Should this say "[0,2]"?
>
> Answer: actually it is [0,0.2]. Thanks for spotting the typo, and we have fixed it in the revised script.
>
> > Re: Line 324 "average amount of the advected smoke simulated by the solver" I don't follow what this means and I can't find the exact details in the supplement.
>
> Answer: we have improved this sentence to “for the optimized parameter, we measure the total amount of smoke simulated by the solver passing through two respective outlets and take their ratio.”
>
> > Re: In the objective defined in line 127, is the right hand side missing integration over x? If not, then there is a different objective for every x location. How are these combined? (Also, less importantly, what does the subscript d refer to?
>
> Answer: The right hand side is not missing integration over x. In fact, the integral should be $L_d[a,X]=\int_{t=t_s}^{t_e}\ell_d[\mathbf{u}(t,\mathbf{x})]dt$ where $\ell_d$ is a functional that maps a continuous function $\mathbf{u}(t,\mathbf{x})$ to a single scalar. The subscript d refers to “design”, as it is close in meaning to “inverse optimization”.
>
> > Re: In Table 1, why is the WEBO5 accumulated error so large if this is the ground truth method?
>
> Answer: the WEBO5 result in the paper is using the ground-truth solver but evolve on the downsampled spatial resolution. Note that the ground-truth data is simulated with spatial resolution of $n_x=200$, but then is downsampled to $n_x=100$, 50 and 40. The larger spatial interval ignores certain information and results in large accumulated error
>
>
> > Re: As someone not familiar with this field, I found the subsection on "Deep learning-based surrogate modeling" reviewed autoregressive methods very clearly. However the 1-sentence summary of "neural operators" was too brief for me to understand. Can you expand on the details of "neural networks that approximate a mapping between infinite-dimensional functions", as this seems non-trivial to do
>
> Answer: due to the space limit we had to make it concise. A typical neural operator M maps $[0,T] \times \mathcal{F} \to \mathcal{F}$, where $\mathcal{F}$ is a (possibly infinite-dimensional) function space. Let’s take Fourier Neural Operator (FNO) for example, regards the data as samples on spatial locations x and time t of the infinite dimensional solution u(t,x), and design the operator as a kernel integral operator that can give value to the output function on (t,x). Certain discretization (in frequency space) is performed to approximate the kernel integral operator. For more information can see Section 2.3 of [1] or FNO paper [1] .
>
> [1] Brandstetter, Johannes, Daniel Worrall, and Max Welling. "Message passing neural PDE solvers." ICLR 2022, arXiv preprint arXiv:2202.03376 (2022).
> [2] Li, Zongyi, et al. "Fourier neural operator for parametric partial differential equations." ICLR 2021, arXiv preprint arXiv:2010.08895 (2020).
>
>
> > Re: Line 107. In what sense is $f$  a partial function?
>
> Answer: Here the function f takes two inputs, first argument is the state $\hat{U}_t$ at each time step, the second argument is the static parameter $p$. When we always give $p$ as a constant in the second argument, then $f(\cdot,p)$ becomes a “partial function” that only takes one argument.

---

> ### Comment · Reviewer_bK7j · 2022-08-08
> **Comment on author response**
>
> Thanks for the response which has resolved all my queries about the submission. As these were only minor points, I have kept my score at 7 (Accept). Congratulations on an excellent paper, which I enjoyed reading and reviewing!

---

### Official Review · Reviewer_Mytm · 2022-07-06

**Rating:** 5
**Confidence:** 2
**Soundness:** 2 fair
**Presentation:** 3 good
**Contribution:** 1 poor

**Summary:**

This paper looks at the problem of expensive time cost when simulating the time evolution of a PDE. Existing methods employ values at different spatial positions at each time step, causing a long simulating time. LE-PDE accelerates by learning a low dimensional latent representation, and evolve the low-dimensional state, rather than the high-dimensional original variable. Optimization is performed upon reconstruction loss of the latent state and the evolution accuracy. Empirical advantages on time cost achieved over previous methods on different PDEs.

**Questions:**

It would be interesting to see a study of when can the proposed LE-PDE method have a reasonable result. Does the reconstruction take extra data to train?
Also, there has been numerous dimension reduction techniques, PCA, VAEs, just to name a few. Why did the author pursue the proposed method rather than making use of the existing ones? Are there any theoretical relationships between the proposed method and the existing ones?

**Limitations:**

The author proposed to accelerate evolution of PDEs by first reducing its dimension.
The idea is intuitive and straight forward, and is proven empirically with different PDEs.
However, a theoretical study on what PDEs this reduction can be efficiently applied to is not conducted, and it is not argued why the proposed reduction method is better than existing ones like VAEs.
The method is flexible, and should be able to be accompanied with existing surrogate models and other method.

**Strengths And Weaknesses:**

Strengths:
Paper demonstrated empirical improvements on both 1D nonlinear PDEs and 2D Navier-Stokes PDE over multiple previous methods.
Model has a relatively flexible structure.

Weaknesses:
Did not study when can PDE states be encoded into a lower dimensional state, and.or how many dimension can be get rid of.
Did not study the number of parameters used against other methods, using an extra encoder and decoder might take extra number of parameters, resulting in unfair advantages over other methods.
Figure 1 is unclear. Specifically, is the box on the left the initial condition and boundary condition of the PDE? The schematic in the middle is also unclear.

---

> ### Author Response · Authors · 2022-08-02
> **Response (3)**
>
>
>
> > Re5: Figure 1 is unclear. Specifically, is the box on the left the initial condition and boundary condition of the PDE? The schematic in the middle is also unclear.
>
> Answer: Thanks for pointing it out! We have updated Fig. 1 in the revised version, with better text notations. Hope that this makes the schematic clearer.
>
> > Re6: It would be interesting to see a study of when can the proposed LE-PDE method have a reasonable result.
>
> Answer: we don’t fully understand the question. Can you clarify it? If we understand correctly, the “reasonable result” means the result measured in the metrics of error and runtime. The main problem we aim to tackle is to speed up the forward simulation and inverse optimization of PDEs. We have shown in extensive experiments (Section 4.1, 4.2, 4.3, and Appendix F) that our LE-PDE achieves significant speedup (which is the main claim of our paper), and as Reviewer bK7j puts it, “the proposed method is always at least competitive with existing methods, and outperforms them in some important examples”.
>
> > Re7: Does the reconstruction take extra data to train?
>
> Answer: No. As is stated in the Section 3.2 of the paper, the model is trained jointly with the three loss terms.

---

> ### Author Response · Authors · 2022-08-02
> **Response (2)**
>
> > Re2: About the novelty of our method
>
> Answer: Here we would like to emphasize the novelty of our method (which is also stated in the Introduction and Related Work sections). The novelty lies in the following aspects: Compared to existing reduced-order modeling methods, (a) we focus on speeding up both simulation and inverse optimization of more general PDEs using expressive NNs, while most existing works focus only on dimension reduction for forward simulation, typically with limitations of the expressivity of the model (e.g. linear projection) and generality (design for narrower applications with domain-specific architecture); (b) we introduce a novel objective that results in better long-term autoregressive rollout. This is also shown in the above additional experiments Table G10, G11, and Table G5, G6 to reviewer jbxg; (c) We demonstrate competitive performance compared to state-of-the-art deep learning-based models for PDEs that evolve on the input space, and demonstrate the scalability of our method to states with millions of cells per time step. Compared to state-of-the-art models that evolve in input space, we clearly demonstrate the effectiveness of our model in speeding up while maintaining competitive accuracy. Compared to [2, 3] that perform inverse optimization on the input space, we instead perform inverse optimization in latent space, which results in speedup and improved accuracy (Section 4.3 of original submission).
>
> [2] K. R. Allen, T. Lopez-Guevara, K. Stachenfeld, A. Sanchez-Gonzalez, P. Battaglia, J. Ham- rick, and T. Pfaff, “Physical design using differentiable learned simulators,” arXiv preprint arXiv:2202.00728, 2022. [3] Q. Zhao, D. B. Lindell, and G. Wetzstein, “Learning to solve pde-constrained inverse problems with graph networks,” International Conference on Machine Learning, 2022.
>
> > Re3: Did not study when can PDE states be encoded into a lower dimensional state, and.or how many dimension can be get rid of.
> However, a theoretical study on what PDEs this reduction can be efficiently applied to is not conducted.
>
> Answer: In fact, we have studied it in Section 4.4 “Ablation study” in the original submission, which we indicated in the main text which points to Table 6, Table 7 and Fig. 6 in Appendix H. As stated in the text in Appendix H of the original submission, we observe that for 1D dataset, “when latent dimension dz is between 16 and 128, the accumulated MSE is near the optimal of 1 ~ 1.1. It reaches minimum at d_z = 64. With larger latent dimensions, e.g. 256 or 512, the error slightly increases, likely due to the overfitting. With a smaller latent dimension (< 8), the accumulated error grows significantly. This shows that the intrinsic dimension of this 1D problem with temporal bundling of S = 25 steps, is somewhere between 4 and 8. Below this intrinsic dimension, the model severely underfits, resulting in huge rollout error.” Please see Appendix H for more details our empirical study of latent dimension for both 1D and 2D datasets. In general, determining the suitable latent dimension of a physical system is an empirical question, which depends on the system’s intrinsic dimension and the model architecture. Theoretically studying it is out of scope of this paper, and may be an interesting future work.
>
> > Re4: Did not study the number of parameters used against other methods, using an extra encoder and decoder might take extra number of parameters, resulting in unfair advantages over other methods
>
> Answer: Thanks for the suggestion! We have performed additional experiments that do extensive hyperparameter search for state-of-the-art FNO model, where the results are provided in Table G1 and G2 in the response to reviewer 1 (jbxg). In Table G3 and Table G4 in the response to reviewer 1, we also provide the number of parameters for our LE-PDE. Please refer to that response for detailed analysis. As shown in the table, our model typically uses much less number of parameters in the latent evolution model (the deciding component in autoregressive rollout), and sometimes more total parameters due to the encoder and decoder. However, as is shown in Table G1 and G2, with a wide hyperparameter search range of FNO, the reported performance of FNO is already near the optimal. In addition, more parameters do not necessarily lead to better performance, since it increases the chance of overfitting. This is shown clearly in Table G1 where FNO with (modes=16, width=128) underperforms FNO with (modes=16, width=64), as well as in Table 6 and 7 in the original submission that increasing the latent dimension (thus total # parameters) too much results in worse performance.

---

> > ### Comment · Reviewer_Mytm · 2022-08-03
> > **Response**
> >
> > Thanks for addressing all my questions, common dimension reduction methods such as VAE indeed perform worse on the PDE evolution task. The newly added ablation study on the number of parameters provides impressive results.
> >
> > I agree that the authors have performed empirical studies on when high-dimensional PDEs can be reduced to lower dimensions. However, it remains unclear which family of PDEs can be applied the proposed method.
> >
> > Despite still lacking theoretical soundness, the empirical results are significant and I will raise my score to 5.

---

> > > ### Author Response · Authors · 2022-08-09
> > > **Response**
> > >
> > > Thanks the reviewers for the response! To give an intuitive answer to your question of "which family of PDEs can be applied the proposed method", we can think of a PDE as a ground-truth model that evolves the ***state*** of a physical system. Typically, the states show more global, dominant features, and can be described by a latent vector with much smaller dimension than the original discretization. Our LE-PDE exploit this ***compressivity of state*** to evolve the system in latent space and achieve speedup, and as long as the PDE does not significantly increase the spatial complexity of the state as it evolves (e.g. developing finer and finer spatial details as in 2-stream instability in of plasma [1]), our method can apply. Most of the PDEs satisfy the above requirements that the state are compressible and does not significantly increase its complexity, so our LE-PDE can be apply to most PDEs. Since any compression of state can incur a possible increase of error (possibly large or small, as the Pareto frontier of "error vs. runtime" and "error vs. #parameter" in Fig. S8 and S9 show for our LE-PDE and FNO), the more important/relevant question is then "what is the tradeoff of error vs. runtime we want for the given PDE", since we can design the encoder of LE-PDE with varying amount of compression. For example, we can design an encoder with minimal compression, so runtime reduction is low but can guarantee to retain low error, or with a much more aggressive compression (like in our 2D and 3D experiments), but can still achieve minimal increase of error. The amount of compression is a hyperparameter which can be optimized and chosen via a validation set. Theoretically studying the best amount of compression that achieves a good tradeoff will be left for an exciting future work. We have also added the above discussion to the end of Appendix J in the revised version.
> > >
> > >
> > > [1] Roberts, K. V., and Herbert L. Berk. "Nonlinear evolution of a two-stream instability." Physical Review Letters 19.6 (1967): 297.

---

> ### Author Response · Authors · 2022-08-02
> **Response (1)**
>
> We thank the reviewer for the comments. Below we address the reviewer’s concerns.
>
> > Re1: There has been numerous dimension reduction techniques, PCA, VAEs, just to name a few. Why did the author pursue the proposed method rather than making use of the existing ones? Are there any theoretical relationships between the proposed method and the existing ones?
>
> Answer: there is an important difference between the PDE simulation setting and the setting used by standard VAE. To learn a model that can simulate the evolution of PDE, whose state can change dramatically during the evolution, we need the learned model to generalize over new states encountered, new boundary and initial conditions, etc. Standard setting for VAE, PCA, in contrast, takes a <em>static<em> setting, which only needs to reduce the dimension but does not need to consider the evolution of the state. Thus, standard dimension reduction techniques cannot apply. Take PCA for example. Looking at the Figure 2 of the paper, we can see that the state at t=0 differs dramatically from state at t=20. The basis of PCA obtained at t=0 will clearly result in a poor performance at t=20.
>
> It is possible to combine existing dimension reduction techniques with our latent evolution model. We have already stated the difference and novelty of our work with prior such works in the “reduced-order modeling” section of Section 2 “Related Work”. In addition, we perform two ablation experiments that explore performing data reduction first and then learn the evolution in latent space: (a) pretrain an autoencoder with states from all time steps, then freeze the autoencoder and train the latent evolution model. This mimics the method in [1]. (b) the encoder and decoder of LE-PDE is replaced with a VAE, first pre-trained with ELBO on all time steps, then freeze the encoder and decoder and train the latent evolution model. All other aspects of the model architecture and training remains the same. The result is shown in the following Table G10 and G11 for the 1D and 2D datasets in Section 4.4 of “Ablation study”.
>
>
> Table G10: Ablation of LE-PDE using pretrained autoencoder or VAE, for 1D dataset E2-50 scenario:
>
> LE-PDE setting | Cumulative error
> :--: | :--:
> LE-PDE (ours) | 1.127
> (a) pretrain autoencoder | 1.952
> (b) pretrained VAE | 1.980
>
>
>
> Table G11: Ablation of LE-PDE using pretrained autoencoder or VAE, for 2D dataset $\nu$=1e-5 scenario
>
> LE-PDE setting | Cumulative error
> :--: | :--:
> LE-PDE (ours) | 0.1861
> (a) pretrain autoencoder | 0.2105
> (b) pretrained VAE | 0.2329
>
>
>
>
> From Table G10 and G11, we see that performing pre-training results in a much worse performance, since the data reduction only focuses on reconstruction, without consideration for <em>which<em> latent state is best used for evolving long-term into the future. On the other hand, our LE-PDE trains the components jointly with a novel objective that not only encourages better reconstruction, but also long-term evolution accuracy both in latent and input space. We also see that VAE as data-reduction performs worse than autoencoder, since the dynamics of the system is deterministic, and having a stochasticity from the VAE does not help.
>
> [1] K. Lee and K. T. Carlberg, “Model reduction of dynamical systems on nonlinear manifolds using deep convolutional autoencoders,” Journal of Computational Physics, vol. 404, p. 108973, 2020

---

### Official Review · Reviewer_MdSE · 2022-07-11

**Rating:** 5
**Confidence:** 4
**Soundness:** 3 good
**Presentation:** 3 good
**Contribution:** 2 fair

**Summary:**

The authors propose a method LE-PDE to efficiently perform forward simulation and inverse optimization of Partial differential equation based models by performing the evolution in latent space. A loss term penalizing 3 terms including the consistency between latent space and the original space is defined, and optimized via backpropagation for the inverse task. Experimental evaluations including an ablation study is provided.


**Questions:**

Please have a look at the weakness mentioned in the strength and weakness section and address these. Overall the idea seems interesting, the authors need to substantiate their claims in light of existing literature and possibly a few more experiments.

**Limitations:**

Yes, the limitations are discussed in the appendix.

**Strengths And Weaknesses:**

Strength:
The authors put together a very clear introduction and motivation for their work, the method is described clearly in an easily reproducible manner.

Weakness:
The authors state that low dimensional representation exists for high dimensional data (line 49-64). Please have a look at Johnson-Lindenstrauss lemma and cite appropriate literature applicable here.

The authors may want to try their experiments on a larger scale. Grid size of 64 is too small, as the authors mention dimensions in the millions/billions in the introduction, at least one experiment should demonstrate the efficacy of the proposed method in such a larger scale where other baseline methods are computationally very very expensive and resource consuming.

Another concern about the proposed method is the applicability of the proposed system in real-world scenarios. Experimental data are often noisy. The authors may want to look at how noise affects the latent space evolution and encoder-decoder performance.

---

> ### Author Response · Authors · 2022-08-02
> **Response (2)**
>
> > Re3: Another concern about the proposed method is the applicability of the proposed system in real-world scenarios. Experimental data are often noisy. The authors may want to look at how noise affects the latent space evolution and encoder-decoder performance.
>
> Answer: Thanks for the suggestion! We have performed additional experiments on how the noise affects the performance, on the representative 1D and 2D dataset used in Section 4.4 “Ablation Study”. Specifically, we add random fixed Gaussian noise to the training, validation and test sets of the dataset, with varying amplitude. The noise is independently added to each feature of the dataset. It is also “fixed” in the sense that once added to the dataset, the noise is freezed and not re-sampled. This mimics the real world setting where random observation noise can corrupt the observation and we never have the ground-truth data to train and evaluate from. Below is the result table:
>
> Table G8: 1D dataset (E2-50 scenario) with varying noise amplitude (the amplitude is the standard deviation of the diagonal Gaussian). The value range of the state u(t,x) is within [-2,2].
>
>
> Noise amplitude | cumulative error
> :--: | :--:
> 0 (default) | 1.127
> 1e-5 | 1.253
> 1e-4 | 1.268
> 1e-3 | 1.456
> 1e-2 | 2.612
> 2e-2 | 4.102
> 5e-2 | 9.228
>
>
> Table G9: 2D dataset ($\nu$=1e-5 scenario) with varying noise amplitude. The value range of the state u(t,x) is within [-2,2]
>
>
>
>
> Noise amplitude | cumulative error
> :--: | :--:
> 0 (default) | 0.1861
> 1e-5 | 0.1880
> 1e-4 | 0.1862
> 1e-3 | 0.1866
> 1e-2 | 0.1897
> 2e-2 | 0.1875
> 5e-2 | 0.1910
> 1e-1 | 0.2012
>
>
> Note that the value range of both datasets are within [-2,2]. From Table G8, we see that LE-PDE’s cumulative error stays excellent (<=1.456) with noise amplitude <= 1e-3, much smaller than state-of-the-art MP-PDE’s error of 1.63 and FNO-PF’s 2.27. Even with noise amplitude of 1e-2, the LE-PDE’s error of 2.612 still remains reasonable.
>
> From Table G9, we see that LE-PDE is quite resilient to noise, with error barely increases for noise amplitude up to 2e-2, and only shows minimal increase at noise level of 1e-1. As a context, U-Net’s error is 0.1982 and TF-Net’s error is 0.2268 (Table 2 in main text).
>
> In summary, in the 1D and 2D datasets, we see that LE-PDE shows good robustness to Gaussian noise, where the performance is reasonable where the ratio of noise amplitude to the value range can go up to 0.25% in 1D and 2.5% in 2D. The smaller robustness in the 1D Burgers’ dataset may be due to that it is a 200-step rollout and the noise may make the model uncertain about the onset of shock formation.
>
>
> Re4: Overall the idea seems interesting, the authors need to substantiate their claims in light of existing literature and possibly a few more experiments.
>
> With the above response, we hope that we have addressed the reviewer’s concerns. The reviewer is also encouraged to look at our response to Reviewer 1 (jbxg) (Table G1 to G6) for additional experiments that further substantiate our claims.

---

> ### Author Response · Authors · 2022-08-02
> **Response (1)**
>
> We thank the reviewer for the feedback. We are glad that the reviewer recognizes the clear introduction, motivation and reproducibility of our work. Below, we address the reviewer’s suggestions/concerns about citation for low-dimensional representation, larger-scale experiment and experiment on influence of noise.
>
> > Re1: The authors state that low dimensional representation exists for high dimensional data (line 49-64). Please have a look at Johnson-Lindenstrauss lemma and cite appropriate literature applicable here.
>
> Answer: Thank you for the suggestion. Indeed, the Johnson-Lindenstrauss lemma proves the existence of a function that can embed given points in a possibly high-dimensional space into a low-dimensional space without distorting more than a factor of $(1 + \epsilon)$. On the one hand, we suspect that the lemma is applicable to PDE systems because PDEs are generally defined on infinite dimensional continuous space and require models to have strong generalizing ability. This is contrary to the lemma that only provides a transductive embedding model. For continuous space, there are also some classical theorems that assure the embedding of smooth manifolds into Euclidean spaces [1]. But this is also transductive. The theoretical study on inductive embedding for PDE systems is unknown in most cases. We believe that the experiments conducted here show some evidence of the model’s ability to inductively embed states and reveal the existence of a dominant global structure that offers the generalization ability as well as the significant speed-up for the model. We will leave theoretical study on the inductive embedding as future research
>
> [1] Whitney H., The self-intersections of a smooth n-manifold in 2n-space. Ann. of Math. (2) 45, (1944). 220-246.
>
>
> > Re2: The authors may want to try their experiments on a larger scale. Grid size of 64 is too small, as the authors mention dimensions in the millions/billions in the introduction, at least one experiment should demonstrate the efficacy of the proposed method in such a larger scale where other baseline methods are computationally very very expensive and resource consuming.
>
> Answer: In fact, we have included a 3D experiment which has 4.19 million cells per time step in the original submission, as stated in the end of Section 4.2 which points to Appendix F. In the original submission, we demonstrated (in Table 5 in page 20 of Appendix) that in a challenging 3D chaotic N-S flow setting, LE-PDE only uses 0.084s to evolve to t=40, where an ablation without latent evolution uses 1.03s, ground-truth solver PhiFlow uses 70.80s on GPU and 1802s on CPU. Thus, LE-PDE achieves a 12.3× speed-up compared to the ablation without latent evolution, and 840× speed-up compared to the ground-truth solver on the same GPU, which is significant.
>
> In addition, per Review 1 (jbxg)’s suggestion, we have also added comparison with current state-of-the-art Fourier Neural Operator (FNO) model in this dataset. The result is in the Table G7 under the Re3 in the response to Reviewer 1 (jbxg). We see that FNO slightly outperforms LE-PDE in terms of long-term rollout error. This is to be expected since LE-PDE uses much less representation dimension (128) than FNO (16.76M). Thus, LE-PDE achieves a much smaller runtime than FNO. This comparison shows that LE-PDE can scale to much larger datasets with millions of dimensions per time step, and achieve significant speedup with minor reduction in performance. We will add this in the revised version of the paper.

---

### Official Review · Reviewer_jbxg · 2022-07-11

**Rating:** 7
**Confidence:** 3
**Soundness:** 3 good
**Presentation:** 3 good
**Contribution:** 3 good

**Summary:**

This paper builds on recent advances in deep learning for physical simulation to reduce the computational complexity of learned surrogate models by avoiding local evolution. Instead of updating the values at each discretized point in space, the authors propose to evolve the model in a global latent space. By encoding both states and boundaries into lower-dimensional latent vectors, the model can evolve dynamics in the latent space and recover the inferred states only when needed. The paper also introduces techniques for training, such as a new consistency loss directly in latent space, a fast way of dealing with inverse optimization problems by backpropagating through time in the latent space, and an annealing technique for boundary optimization. Experiments in various PDE settings demonstrate that the proposed model, LE-PDE (Latent-Evolution of PDEs) compares competitively against state-of-the-art methods while requiring a fraction of the other methods’ computational resources.

**Questions:**

1. GNN+MLPs as a future direction, but it is unclear to me whether this could be possible. How would message passing be realized in a global latent space, though? Could we still use the same latent space even if we added particles or mesh points?

2. About the boundary annealing, what is meant by $\beta$ being a “temperature hyperparameter”? This notation is also not introduced in the main text.

**Limitations:**

See above. As stated in the main text, no major negative societal impact is to be expected from this work.

**Strengths And Weaknesses:**

### Strenghts

The paper is appropriately placed in the current growing literature on scientific machine learning to create a novel fast model for tackling complex problems in the realm of PDEs. The exposition for the motivation is written crisply and is generally easy to follow. The experiments are performed in challenging settings, such as boundary control with the 2D Navier Stokes equations. The results show that very large neural networks are not always necessary to represent complex physics and that “simple” latent MLPs with few parameters can learn such physics evolutions thus considerably lowering computational requirements, which is significant in deep learning for simulation.

### Weaknesses

Although the speedups of LE-PDE are considerable, it is unclear how the proposed approach would compare against other models with a similar number of parameters. What if the FNO had fewer parameters, comparable to LE-PDE? Would it still be better than the proposed approach? In other words, it is unclear whether the method is *Pareto efficient* or not since there is no Pareto plot considering an extensive hyperparameter search for both LE-PDE and close competitors such as the FNO.

About the novelty: there is a possibly missing reference [1] (although not published in conference proceedings), in which the authors evolve a model in the latent space (see Figure 1 and Section 3.1 of the reference), and the main idea of the paper is very similar to the presented one. How does this work compare against it?

The experimental section in the Appendix about the 3D extension of the method (3D Navier Stokes) lacks some experimental details and evaluation. The learned model is not compared with other deep learning approaches, so it is difficult to assess how the proposed method would perform against other models (except for the same version of the model without the global latent space evolution). While the comparison in the main text is against other deep learning approaches, here it is against the ground truth solver which is known to be generally much slower than deep learning approaches. Moreover, there is no report on statistical errors but only a qualitative plot (that also shows noticeable artifacts).

[1] Sanchez A, Kochkov D, Smith JA, Brenner M, Battaglia P, Pfaff TJ. Learning latent field dynamics of PDEs,  Third Workshop on Machine Learning and the Physical Sciences (NeurIPS 2020)

### Minor comments

These are primarily typos and do not influence my score:

- Line 235-236: “Burgur’s Equation”
- Line 355: “sweep spot”
- Line 707 (Appendix): "The details of the dataset has already given in […]”
- Line 748: “To explore how LE-PDE to larger scale turbulent dynamics a […]”

---

> ### Author Response · Authors · 2022-08-02
> **Official Response (4)**
>
> > Re3: The experimental section in the Appendix about the 3D extension of the method (3D Navier Stokes) lacks some experimental details and evaluation. The learned model is not compared with other deep learning approaches, so it is difficult to assess how the proposed method would perform against other models
>
> Answer: Thanks for the comment. We perform an additional experiment on the 3D dataset that compares with FNO, the state-of-the-art model. We see from the table below G5 that FNO slightly outperforms LE-PDE in terms of long-term rollout error. This is to be expected since LE-PDE uses much less representation dimension (128) than FNO (16.76M).  In the table, we see that our LE-PDE uses much less number of parameters to evolve autoregressively than FNO. The most parameters of LE-PDE are mainly in the encoder and decoder. Thus, LE-PDE achieves a much smaller runtime than FNO to evolve to t=40. The results are updated in the Appendix F in the revised Appendix.
>
> Table G7: 3D experiment that compare our model with FNO
>
> LE-PDE setting | error at t=40 | runtime (s) | # parameters | # parameters of evolution model
> :--: | :--: | :--: | :--: | :--:
> FNO | 0.1695  | 7.00 | 3281864 | 3281864
> LE-PDE (ours) | 0.1947 | 0.084 | 65003120 | 83072
>
> We have updated the table 5 in Appendix F in the revised version.
>
>
>
> > Re4: Question 1: GNN+MLPs as a future direction, but it is unclear to me whether this could be possible. How would message passing be realized in a global latent space, though? Could we still use the same latent space even if we added particles or mesh points?
>
> Answer: By GNN+MLP as a future direction, we mean that similar to CNN+MLP scenario, the GNN can have a few message passing layers with local graph pooling that reduces the number of nodes but increases the feature dimension. Then a global pooling is performed on the GNN that results in a single vector, which feeds into an MLP that produces the latent representation. This architecture would generalize the CNN + MLP architecture. If the local graph pooling and global graph pooling is invariant to the permutation of nodes, then this architecture can adapt to added particles or mesh points at test time. This is an interesting direction for future research.
>
> > Re5: Question 2: About the boundary annealing, what is meant by β being a “temperature hyperparameter”? This notation is also not introduced in the main text.
>
> Answer: First, we apologize for causing confusion by not clarifying the notation of the temperature parameter in the main text. Typically, boundary information in PDE systems is provided as a binary mask which indicates which cells are outside of the simulation domain. This discreteness actually makes the inverse boundary optimization difficult because it is not possible to perform backpropagation through discrete variables. We therefore introduced a continuous boundary mask that interpolates the discrete boundary mask and continuous variables. The parameter β is set in this continuous mask and plays a role in controlling the degree of continuity; if β is small enough, then the mask accurately approximates the discrete boundary mask. Especially, when we perform the inverse boundary optimization, we simultaneously run an annealing technique where large β of the early stage of iteration gradually  becomes small. We call β the “temperature” hyperparameter because the parameter gets hard to be updated as β gets “cooler” (i.e., smaller). We include the details on the formulation in Appendix B. We hope that the explanation above and the appendix resolve your question.
>
> > Re6: Minor comments on typos:
>
> Answer: Thanks for spotting the typos. We have fixed the typos in the revised version.
>
>
> With the above additional results, we hope that we have resolved the reviewer’s concerns and have strengthened the paper.

---

> > ### Comment · Reviewer_jbxg · 2022-08-07
> > **Response**
> >
> > I would like to thank the authors for providing additional experiments and updating the manuscript, especially for providing more results for the Pareto comparison, the 3D experiment details, and additional results compared to LFM.
> >
> > However, I have some questions regarding the updated experimental results, in particular regarding the number of parameters and training times.
> >
> > 1. Regarding the Pareto front comparison, given that the results are in a table format (separated for both FNO and LE-PDE), it may be a bit hard to directly compare the results, and I would advise making, perhaps, a plot (e.g. error vs # parameters, error vs runtime) to better understand the differences. For example, looking at an extract of the table for the 2D table (best LE-PDE model vs default FNO):
> >
> > | Model |  Error   |  Runtime |  Parameter count |
> > |-------------|------------------|-------------------|-------------------|
> > | FNO (modes=12, width=20 (default setting))        | 	$0.1745$  |	$42.7 \pm10.9$	|    $465,717$ |
> > | LE-PDE (d_z=256)|	$0.1861$  | $14.8 \pm 1.1$  | $3,384,944$  |
> >
> > While LE-PDE is around 3 times faster, the number of parameters is several times the ones of the FNO. In many cases, the number of parameters can be more than 10 times the ones required by the FNO. While I understand the latent parameters are less, it seems this number is due to the learned encoder and decoders. My question then is, how long is the training time for FNO vs LE-PDE? Given the high number of parameters, it looks like it should be several times as much, and I don't recall seeing such a metric in the main text and rebuttals.
> >
> > 2. Similar to the question before, the 3D case also shows a substantial increase in the number of total parameters for the LE-PDE (~ 20$\times$ as many).
> >
> > To summarize, my main concern is about the impact on memory requirements of the proposed model as well as training times.

---

> > > ### Author Response · Authors · 2022-08-08
> > > **Response**
> > >
> > > >  Regarding the Pareto front comparison, given that the results are in a table format (separated for both FNO and LE-PDE), it may be a bit hard to directly compare the results, and I would advise making, perhaps, a plot (e.g. error vs # parameters, error vs runtime) to better understand the differences. For example, looking at an extract of the table for the 2D table (best LE-PDE model vs default FNO)
> > >
> > > **Answer:** Thank you for the suggestion! We created plots that present error v.s. # parameters and error v.s. runtime trade-offs from the tables G1, G2, G3 and G4. Please see Figure S8 and S9 in Appendix J (pp. 26-27) in the newly revised version of the paper. We hope that the plots could provide a clear understanding of the differences between the models. In summary, for the error vs. runtime plot, LE-PDE Pareto-dominates FNO in 1D (Fig. S8(a)), and have much smaller runtime and comparable error w.r.t. FNO in 2D (Fig. S8(b)). For the error vs. #parameter plot, LE-PDE with evolution model typically has less parameters than FNO, which in turn also have less parameters than LE-PDE with full model (Fig. S9(a)(b)).
> > >
> > > > My question then is, how long is the training time for FNO vs LE-PDE? Similar to the question before, the 3D case also shows a substantial increase in the number of total parameters for the LE-PDE (~ 20 as many). To summarize, my main concern is about the impact on memory requirements of the proposed model as well as training times.
> > >
> > >
> > > **Answer:** Thank you for the additional comments. This is a very good point. Here, we present an augmented table G12 as an updated table of G7 in the following (we also updated the Table 5 in Appendix F, page 20, in the newly revised version); we add two metrics, training time per epoch and GPU memory usage during training, which are impacted by the parameters of the evolution models and other factors (e.g. architecture):
> > >
> > > Table G12: Comparison of LE-PDE with baseline on runtime and representation dimension, in the 3D Navier-Stokes flow. The runtime is to predict the state at t = 40.
> > >
> > > LE-PDE setting | error at t=40 | runtime for rollout |  # parameters | # parameters for LE-PDE | training time (min) per epoch  | memory usage (MiB)
> > > :--: | :--: | :--: | :--: | :--: | :--: | :--:
> > > FNO with 2-step loss | 0.1695 | 7.00 |  3281864 | 3281864  |  102 | 25147
> > > FNO with 1-step loss | 0.3215 | 7.00 |  3281864 | 3281864  | 58 | 24891
> > > LE-PDE (ours) | 0.1947 | 0.084 | 65003120 | 83072  | 65 | 25595
> > >
> > > The “FNO with 2-step loss” is trained with 2-step rollout. We also added results for FNO trained with single-step loss in the Table. Comparing “LE-PDE (ours)” with “FNO with 2-step loss”, we see that although ours has much more total parameters, the training time is smaller, and memory usage is comparable. Concretely,
> > >
> > > **memory usage:** The reason why FNO has similar memory usage as ours albeit less parameter is the following: the default FNO model has 4 SpectralConv3d layers, 4 Conv3d layers and 2 dense layers, without spatial compression, and during the training, each layer needs to retain the intermediate layer activations (during forward) and gradients during backpropagation. Therefore, the intermediate layer activations and gradients of 4.2 million cells are stored for 4+4+2 layers. In contrast, our LE-PDE’s encoder has 5 layers of Conv3d, each with a compression rate of 4, resulting in much smaller requirement for storing intermediate layer activations and gradients.
> > >
> > > **Training time:** The smaller training time of our LE-PDE compared with “FNO with 2-step loss” is mainly due to LE-PDE’s smaller runtime for rollout (third column), which is an inner loop of training. We also see that “FNO with 1-step loss” halves the training time compared to “FNO with 2-step loss”, since the latter does not need to rollout for 2 steps during training as an inner loop. However, “FNO with 1-step loss” has a much larger error.
> > >
> > > In summary, we see that in 3D large-scale experiment, although the full parameters of our LE-PDE can be larger than FNO, the memory usage and training time is roughly comparable. In 1D and 2D, we observe similar memory usage between FNO and our LE-PDE, and FNO’s training time is slightly less.

---

> > > > ### Comment · Reviewer_jbxg · 2022-08-08
> > > > **Response**
> > > >
> > > > Thank you for your answers.
> > > >
> > > > It is still a bit unclear about the extent to which the parameters for the encoder and decoder actually contribute to learning the PDE since at this point I would assume that they do much of the work in compressing its representation - they would evolve the dynamics only, while they don't really compress all of the PDE itself. In other words, while a small MLP may not be able to fully compress dynamics (as pointed out by other reviewers), it may be able to evolve them in latent space (hence the title of the paper), while encoders and decoders would have most of the parameters tasked with compression and reconstruction.
> > > >
> > > > In summary, although the number of parameters of LE-PDE is noticeably larger, given your explanation and results on training times and memory usage along with the new Pareto plots, my major concerns are resolved.
> > > >
> > > > Thus, given the contributions of the paper as well as the willingness of the authors' to provide explanations and a considerable amount of additional experimental results, I would recommend the paper for acceptance and am raising my score to 7.

---

> ### Author Response · Authors · 2022-08-02
> **Official response (3)**
>
> > Re2: About the novelty: there is a possibly missing reference [1] (although not published in conference proceedings), in which the authors evolve a model in the latent space (see Figure 1 and Section 3.1 of the reference), and the main idea of the paper is very similar to the presented one. How does this work compare against it?
>
> Answer: Thanks for pointing us to the Latent Field Model (LFM, reference [1]), which we have added the citation in the revised version. Our LE-PDE differs from LFM in three major aspects: (1) local vs. global representation: to improve speed, LE-PDE requires an MLP in the encoder and decoder, which makes the latent representation global. In comparison, LFM requires the full architecture to be local, so has no MLP in the encoder and decoder. (2) Objective: we introduce novel learning objective, which encourage the matching of values in both the input and latent space after long-term rollout. In comparison, LFM’s objective encourages the matching of time-derivative in both the latent space and in input space, connected by the Jacobian of encoder or decoder. The LFM objective is good with very small time intervals Δt, but will become imprecise with larger time intervals Δt: with larger time intervals where the states change dramatically, the Jacobian of the encoder or decoder w.r.t. input may also change and we may not be able to use the Jacobian at time t to approximate the Jacobian across [t, t+Δt]. On the other hand, since LE-PDE’s objective encourages matching of values, it is valid for even large intervals. (3) Experiment evaluation: LFM is only evaluated in a 1D PDE, while we evaluate on a 1D PDE dataset, a challenging 2D dataset, and a more challenging 3D datasets, as well as inverse optimization problem, demonstrating the wide applicability and scalability of our method.
>
> We also perform additional experiments to compare our LE-PDE with LFM, in the representative 1D and 2D datasets in Section 4.4. As noted above, LFM differs with LE-PDE in (1) architecture and (2), therefore, we perform the ablation study where we (a) remove MLP in our model (b) use LFM objective, but maintain MLP (c) full LFM: remove MLP, use LFM objective, while all other aspects of training is kept the same. We use PyTorch’s jvp function in autograd to compute the Jacobian-vector product and carefully make sure that our implementation is correct. Below is the comparison table. From the tables, we see that without MLP, it actually results in worse performance (ablation (a)), and with LFM objective, the error is larger, likely due to that the dataset are quite chaotic and LFM may not adapt to the large time range in these datasets. From the below tables, we see that without MLP, it actually results in worse performance (ablation (a)), and with LFM objective, the error is larger, likely due to that the dataset are quite chaotic and LFM may not adapt to the large time range in these datasets.
>
> Table G5: Comparison of LE-PDE with LFM, for 1D dataset E2-50 scenario:
>
> LE-PDE setting | cumulative error | runtime (full) (ms) | runtime (evolution) (ms) | # parameters | # parameters for latent evolution model
> :--: | :--: | :--: | :--: | :--: | :--:
> LE-PDE (ours) | 1.127 | 14.9 $\pm$ 1.1 | 6.0 $\pm$ 0.4 | 1630976 | 82944
> (a) without MLP | 7.930 | 17.2 $\pm$ 6.0 | 8.3 $\pm$ 0.4 | 2730368 | 1580544
> (b) with LFM objective | 58.85 | 15.7 $\pm$ 1.5 | 6.5 $\pm$ 0.6 | 1630976 | 82944
> (c) full LFM: without MLP, with LFM objective | 26.12 | 15.7 $\pm$ 1.3 | 8.4 $\pm$ 0.7 | 2730368 | 1580544
>
> Table G6: Comparison of LE-PDE with LFM, for 2D dataset $\nu$=1e-5 scenario
>
> LE-PDE setting | cumulative error | runtime (full) (ms) | runtime (evolution) (ms) | # parameters | # parameters for latent evolution model
> :--: | :--: | :--: | :--: | :--: | :--:
> LE-PDE (ours) | 0.1861 | 14.8 $\pm$ 1.1 | 5.8 $\pm$ 0.4 | 3384944 | 328960
> (a) without MLP | 0.2120 | 16.6 $\pm$ 2.2 | 9.2 $\pm$ 0.8 | 2126960 | 1181184
> (b) with LFM objective | 0.4530 | 15.8 $\pm$ 2.3 | 6.2 $\pm$ 0.6 | 3384944 | 328960
> (c) full LFM: without MLP, with LFM objective | 0.6315 | 16.2 $\pm$ 1.9 | 9.1 $\pm$ 0.4 | 2126960 | 1181184
>
> We have added the comparison to the Appendix K of the revised version.

---

> ### Author Response · Authors · 2022-08-02
> **Official Response (2)**
>
> For comparison, here we also provide augmented table of our LE-PDE by varying the latent dimension (d_z), for 1D dataset (Table G3) and 2D dataset (Table G4). This includes results in Table 6, 7 in Appendix H but also provides additional information about the number of parameters. Note that we provide both total number of parameters (second last column) and # parameters for latent evolution model (last column). The latter is also a good indicator since during long-term evolution, the latent evolution model is autoregressively applies while encoder and decoder is only applied once. So the latent evolution model is the deciding component of the long-term evolution accuracy and runtime.
>
> Table G3: 1D dataset (E2-50 scenario) with LE-PDE:
>
>
> LE-PDE setting | cumulative error |  runtime** (full) (ms) | runtime (evolution) (ms) | # parameters | # parameters for latent evolution model
> :--: | :--: | :--: | :--: | :--: | :--:
> d_z=512 | 2.778 | 16.3 $\pm$ 2.6 | 6.7 $\pm$ 1.0 | 4043648 | 1314816
> d_z=256 | 2.186 | 15.0 $\pm$ 0.8 | 6.1 $\pm$ 0.3 | 2271360 | 329728
> d_z=128 | 1.127 | 14.9 $\pm$ 1.1 | 6.0 $\pm$ 0.4 | 1630976 | 82944
> d_z=64 | 0.994 | 14.4 $\pm$ 1.0 | 5.7 $\pm$ 0.3 | 1372224 | 20992
> d_z=32 | 1.048 | 14.5 $\pm$ 0.8 | 5.8 $\pm$ 0.4 | 1258208 | 5376
> d_z=16 | 1.041 | 14.1 $\pm$ 0.9 | 5.8 $\pm$ 0.4 | 1205040 | 1408
> d_z=8 | 21.03 | 14.0 $\pm$ 0.7 | 5.6 $\pm$ 0.2 | 1179416 | 384
> d_z=4 | 205.09 | 13.9 $\pm$ 0.5 | 5.7 $\pm$ 0.3 | 1166844 | 112
>
> ** The runtime value here slightly differs from that in Table 6 in paper due to that the GPU machine was busy at the time of running. Here we make sure the 4 current tables (Table G1 to G4) are run on the same machine and same environment, so the comparison is fair.
>
> Table G4: 2D dataset ($\nu$=1e-5 scenario) for LE-PDE:
>
> LE-PDE setting | cumulative error | runtime (full) (ms) | runtime (evolution) (ms) | # parameters | # parameters for latent evolution model
> :--: | :--: | :--: | :--: | :--: | :--:
> d_z=512 | 0.1930 | 16.2 $\pm$ 1.1 | 6.8 $\pm$ 0.7 | 6467184 | 1313280
> d_z=256 | 0.1861 | 14.8 $\pm$ 1.1 | 5.8 $\pm$ 0.4 | 3384944 | 328960
> d_z=128 | 0.2064 | 14.8 $\pm$ 0.5 | 5.9 $\pm$ 0.4 | 2089584 | 82560
> d_z=64 | 0.2252 | 14.7 $\pm$ 0.7 | 6.0 $\pm$ 0.7 | 1503344 | 20800
> d_z=32 | 0.2315 | 15.0 $\pm$ 2.1 | 5.9 $\pm$ 0.5 | 1225584 | 5280
> d_z=16 | 0.2236 | 14.2 $\pm$ 1.3 | 5.8 $\pm$ 0.6 | 1090544 | 1360
> d_z=8 | 0.3539 | 14.3 $\pm$ 0.6 | 5.7 $\pm$ 0.3 | 1023984 | 360
> d_z=4 | 0.6353 | 14.2 $\pm$ 0.5 | 5.7 $\pm$ 0.2 | 990944 | 100
>
>
>
> From the comparison, we see that:
>
> For 1D dataset, LE-PDE Pareto-dominates FNO in error vs. runtime plot. FNO’s best cumulative error is 2.240, and runtime is above 17.9ms, over the full hyperparameters combinations (# parameter varying from 6953 to 1.4M). In comparison, our LE-PDE achieves much better error and runtime over a wide parameter range: for d_z from 16 to 64, LE-PDE’s cumulative error <= 1.05, runtime <=14.5ms, latent runtime <=5.8ms, (which uses 1408 to 82944 number of parameters for latent evolution model, and ~1.2-1.4M total parameters).
>
> For 2D dataset, FNO’s cumulative error is slightly better than LE-PDE, but its runtime is significantly larger. Concretely, the best FNO achieves an error of 0.1454 while the best LE-PDE’s error is 0.1861. FNO’s runtime is above 40ms, while LE-PDE’s runtime is generally below 15ms and latent evolution runtime is below 6ms. LE-PDE uses larger total number of parameters but much less # parameters for latent evolution model.
>
> We will add the above tables to the Appendix of the paper and also add a Pareto plot of error vs. runtime for both models.

---

> ### Author Response · Authors · 2022-08-02
> **Official Response (1)**
>
> We thank the reviewer for the positive and detailed feedback. We are glad that the reviewer recognizes the significance, clarity and challenging experimental setting of our work. Below, we address the reviewer’s points on Pareto efficiency, comparison with one existing work, and 3D experiment.
>
> > Re1: It is unclear how the proposed approach would compare against other models with a similar number of parameters. In other words, it is unclear whether the method is Pareto efficient or not since there is no Pareto plot considering an extensive hyperparameter search for both LE-PDE and close competitors such as the FNO.
>
> Answer: This is a good point. Following the reviewer’s suggestion, we have performed additional experiments that do extensive hyperparameter search for FNO, on the two representative settings of the 1D and 2D datasets used in Section 4.4 “Ablation Study”, which we provide the result table below. We would also like to point the reviewer to the existing ablation study of our LE-PDE in Appendix H, where we have performed extensive hyperparameter search that varies the latent dimension of our LE-PDE, which is the most important hyperparameter in the LE-PDE architecture, and determines the number of parameters in the latent evolution model and the runtime. As a summarization of results, For 1D dataset, LE-PDE Pareto-dominates FNO in error vs. runtime plot. For 2D dataset, FNO’s cumulative error is slightly better than LE-PDE, but its runtime is significantly larger.
>
> In the following, we present two tables (Table G1, Table G2) for FNO hyperparameter search on 1D and 2D datasets in Section 4.4, respectively. For FNO, the most important hyperparameters are the “modes”, which denotes the number of Fourier frequency modes, and “width”, which denotes the channel size for the convolution layer in the FNO. We vary both values starting from the default setting:
>
> Table G1: 1D dataset (E2-50 scenario) with FNO:
>
> | FNO setting* | cumulative error | runtime (full) (ms) | # parameters |
> | :--: | :--: | :--: | :--: |
> | modes=16, width=64 (default setting) | 2.379 | 21.2 $\pm$ 6.9 | 292249 |
> | modes=16, width=128 | 3.107 | 21.7 $\pm$ 4.3 | 1138201 |
> | modes=16, width=32 | 2.695 | 22.1 $\pm$ 7.4 | 78169 |
> | modes=16, width=16 | 2.755 | 21.0 $\pm$ 5.7 | 23353 |
> | modes=16, width=8 | 4.992 | 17.9 $\pm$ 1.2 | 9001 |
> | modes=20, width=128 | 2.804 | 20.9 $\pm$ 1.1 | 1400345 |
> | modes=20, width=64 | 2.626 | 19.3 $\pm$ 0.9 | 357785 |
> | modes=12, width=64 | 2.899 | 19.6 $\pm$ 2.2 | 226713 |
> | modes=8, width=64 | 2.240 | 19.7 $\pm$ 1.3 | 161177 |
> | modes=4, width=64 | 2.326 | 19.2 $\pm$ 0.9 | 95641 |
> | modes=8, width=32 | 2.366 | 18.2 $\pm$ 1.0 | 45401 |
> | modes=8, width=16 | 2.505 | 18.1 $\pm$ 1.2 | 15161 |
> | modes=8, width=8 | 5.817 | 18.4 $\pm$ 1.2 | 6953 |
>
> *Here the FNO follows the FNO-PF setting implemented in the MP-PDE paper (Brandstetter et al. 2022). The runtime is the average of 100 runs.
>
>
> Table G2:2D dataset ($\nu$=1e-5 scenario) for FNO:
>
>
> FNO setting | cumulative L2 error | runtime (full) (ms) | # parameters
> :--: | :--: | :--: | :--:
> modes=12, width=20 (default setting) | 0.1745 | 42.7 $\pm$ 10.9 | 465717
> modes=12, width=40 | 0.1454 | 42.7 $\pm$ 4.2 | 1855977
> modes=12, width=10 | 0.2016 | 40.3 $\pm$ 5.4 | 117387
> modes=12, width=5 | 0.2398 | 45.5 $\pm$ 7.4 | 29922
> modes=16, width=20 | 0.1710 | 43.7 $\pm$ 4.2 | 824117
> modes=8, width=20 | 0.1770 | 43.1 $\pm$ 3.1 | 209717
> modes=4, width=20 | 0.1997 | 43.2 $\pm$ 4.8 | 56117
> modes=8, width=10 | 0.2109 | 42.2 $\pm$ 4.8 | 53387
> modes=8, width=5 | 0.2415 | 43.3 $\pm$ 4.3 | 13922

---

### Author Response · Authors · 2022-08-02
**General Response**

General response:
We thank the reviewers for their thorough and constructive comments. Reviewers agree that our method is simple, flexible, and shows significant speed-up.  Based on reviewers’ valuable feedback, we have conducted a number of additional experiments, which resolve the reviewers’ concerns. We have also updated the paper and Appendix in the revised version. The major additional experiments and improvements are as follows:

1. To address the concern about Pareto efficiency, we do additional experiments that perform extensive hyperparameter search for FNO on 1D (Table G1) and 2D (Table G2) datasets, and also provide the number of parameters of our LE-PDE with varying latent dimension, for 1D (Table G3) and 2D dataset (Table G4). Experiments show that LE-PDE pareto-dominates FNO in error vs. runtime plot in most of the cases.  They are under the Re1 for Reviewer jbxg. Also added appendix J in the revised version.

2. To complement LE-PDE’s result on 3D dataset with 4.19 million cells per time step, we perform an additional experiment that runs FNO on this 3D dataset. The result table is provided in Table G7 under the Re3 for Reviewer jbxg, which also addresses one of Reviewer MdSE’s concerns. The experiment shows that our model achieves significant speed-up while keeping competitive long-term rollout error for the baseline. Detailed experimental results are below.  We also updated Appendix F.

3. We additionally compare with Latent Field Model (LFM), a model that also uses latent evolution, but differs from our work in the architecture, objective and experimental evaluation. The result table is provided in Table G5 and G6 under the Re2 for Reviewer jbxg. We see that with LFM objective, the error is larger. Detailed are added at Appendix K.

4. We run an additional experiment that explores how noise influences the performance of our model. It is provided in Table G8 and G9 under the Re3 for reviewer MdSE. We see that LE-PDE shows good robustness to Gaussian noise, where the performance is reasonable where the ratio of noise amplitude to the value range can go up to 0.25% in 1D and 2.5% in 2D. Details are added at Appendix L.

5. We run additional experiments that compare our LE-PDE with the model that pretrains an autoencoder or VAE for dimension reduction, then freeze the autoencoder and train the latent evolution model. Results are shown in Table G10 and G11 for reviewer Mytm. We see that performing pre-training results in a much worse performance. Details are added at Appendix M.

Other concerns are individually addressed in the response to each reviewer. We also emphasize the novelty of our method in the response to reviewer Mytm.

In summary, through extensive experiments in original submission (in the main text and Appendix) and additional experiments in the following response, we show general applicability and scalability of LE-PDE to different scenarios, and its relative strengths compared with current state-of-the-art models and baselines. We hope that our work makes a useful step to help speed up the forward simulation and inverse optimization of PDEs, pivotal in science and engineering.

---

### Meta-Review · Area_Chair_SwbH · 2022-08-31

**Recommendation:** Accept
**Confidence:** Certain

**Metareview:**

The paper presents a new method for accelerating the simulation and inverse optimization of partial differential equations (PDEs) of large-scale systems. The proposed approach learns the evolution of dynamics in a “global” latent space (i.e., with fixed dimensionality). The reviewers agree the proposed approach is novel and empirically competitive. issues regarding experiments have largely been addressed by the authors in their rebuttal. Their authors are expected to add some extended discussion (if possible) on (theoretical) properties of PDEs where their approach is expected to succeed. Some of the reviewers increased their scores after the rebuttal period.

**Award:**

No

---

### Decision · Program_Chairs · 2022-09-14

Accept